# UNDERSTANDING THE COVARIANCE STRUCTURE OF CONVOLUTIONAL FILTERS

**Asher Trockman**[1]**, Devin Willmott**[2]**, J. Zico Kolter**[12]
[1]Carnegie Mellon University and [2]Bosch Center for AI
Correspondence to: `ashert@cs.cmu.edu`

## ABSTRACT

Neural network weights are typically initialized at random from univariate distributions, controlling just the variance of individual weights even in highly-structured operations like convolutions. Recent ViT-inspired convolutional networks such as ConvMixer and ConvNeXt use large-kernel depthwise convolutions whose learned filters have notable structure; this presents an opportunity to study their empirical covariances. In this work, we first observe that such learned filters have highly-structured covariance matrices, and moreover, we find that covariances calculated from a small network may be used to effectively initialize a variety of larger networks of different depths, widths, patch sizes, and kernel sizes, indicating a degree of model-independence to the covariance structure. Motivated by this finding, we then propose a learning-free *multivariate* initialization scheme for convolutional filters using a simple, closed-form construction of their covariance. Models using our initialization outperform those using traditional univariate initializations, and typically meet or exceed the performance of those initialized from the covariances of learned filters; in some cases, this improvement can be achieved *without training the depthwise convolutional filters at all*. Our code is available at `https://github.com/locuslab/convcov`.

## 1 INTRODUCTION

Early work in deep learning for vision demonstrated that the convolutional filters in trained neural networks are often highly-structured, in some cases being qualitatively similar to filters known from classical computer vision (Krizhevsky et al., 2017). However, for many years it became standard to replace large-filter convolutions with stacked small-filter convolutions, which have less room for any notable amount of structure. But in the past year, this trend has changed with inspiration from the long-range spatial mixing abilities of vision transformers. Some of the most prominent new convolutional neural networks, such as ConvNeXt and ConvMixer, once again use large-filter convolutions. These new models also completely separate the processing of the channel and spatial dimensions, meaning that the now-single-channel filters are, in some sense, more independent from each other than in previous models such as ResNets. This presents an opportunity to investigate the structure of convolutional filters.

In particular, we seek to understand the *statistical structure* of convolutional filters, with the goal of more effectively initializing them. Most initialization strategies for neural networks focus simply on controlling the *variance* of weights, as in Kaiming (He et al., 2015) and Xavier (Glorot & Bengio, 2010) initialization, which neglect the fact that many layers in neural networks are highly-structured, with interdependencies between weights, particularly after training. Consequently, we study the *covariance* matrices of the parameters of convolutional filters, which we find to have a large degree of perhaps-interpretable structure. We observe that the covariance of filters calculated from pre-trained models can be used to effectively initialize new convolutions by sampling filters from the corresponding multivariate Gaussian distribution.

We then propose a closed-form and completely learning-free construction of covariance matrices for randomly initializing convolutional filters from Gaussian distributions. Our initialization is *highly effective*, especially for larger filters, deeper models, and shorter training times; it usually outperforms both standard uniform initialization techniques *and* our baseline technique of initializing by

sampling from the distributions of pre-trained filters, both in terms of final accuracy and time-to-convergence. Models using our initialization often see gains of over 1% accuracy on CIFAR-10 and short-training ImageNet classification; it also leads to small but significant performance gains on full-scale, $\approx 80\%$-accuracy ImageNet training. Indeed, in some cases our initialization works so well that it outperforms uniform initialization *even when the filters aren't trained at all*. And our initialization is almost completely *free to compute*.

**Related work** Saxe et al. (2013) proposed to replace random *i.i.d.* Gaussian weights with random orthogonal matrices, a constraint in which weights depend on each other and are thus, in some sense, "multivariate"; Xiao et al. (2018) also proposed an orthogonal initialization for convolutions. Similarly to these works, our initialization greatly improves the trainability of deep (depthwise) convolutional networks, but is much simpler and constraint-free, being just a random sample from a multivariate Gaussian distribution. Zhang et al. (2022) suggests that the main purpose of pre-training may be to find a good initialization, and crafts a *mimicking initialization* based on observed, desirable information transfer patterns. We similarly initialize convolutional filters to be closer to those found in pre-trained models, but do so in a completely random and simpler manner. Romero et al. (2021) proposes an analytic parameterization of variable-size convolutions, based in part on Gaussian filters; while our covariance construction is also analytic and built upon Gaussian filters, we use them to specify the *distribution* of filters.

Our contribution is most advantageous for large-filter convolutions, which have become prevalent in recent work: ConvNeXt (Liu et al., 2022b) uses $7 \times 7$ convolutions, and ConvMixer (Trockman & Kolter, 2022) uses $9 \times 9$; taking the trend a step further, Ding et al. (2022) uses $31 \times 31$, and Liu et al. (2022a) uses $51 \times 51$ sparse convolutions. Many other works argue for large-filter convolutions (Wang et al., 2022; Chen et al., 2022; Han et al., 2021).

**Preliminaries** This work is concerned with depthwise convolutional filters, each of which is parametrized by a $k \times k$ matrix, where $k$ (generally odd) denotes the filter's size. Our aim is to study distributions that arise from convolutional filters in pretrained networks, and to explore properties of distributions whose samples produce strong initial parameters for convolutional layers. More specifically, we hope to understand the covariance among pairs of filter parameters for fixed filter size $k$. This is intuitively expressed as a covariance matrix $\Sigma \in \mathbb{R}^{k^2 \times k^2}$ with block structure: $\Sigma$ has $k \times k$ blocks, where each block $[\Sigma_{i,j}] \in \mathbb{R}^{k \times k}$ corresponds to the covariance between filter pixel $i, j$ and all other $k^2 - 1$ filter pixels. That is, $[\Sigma_{i,j}]_{\ell,m} = [\Sigma_{\ell,m}]_{i,j}$ gives the covariance of pixels $i, j$ and $\ell, m$.

In practice, we restrict our study to multivariate Gaussian distributions, which by convention are considered as distributions over $n$-dimensional *vectors* rather than matrices, where the distribution $\mathcal{N}(\mu, \Sigma')$ has a covariance matrix $\Sigma' \in \mathbb{S}_+^n$ where $\Sigma'_{i,j} = \Sigma'_{j,i}$ represents the covariance between vector elements $i$ and $j$. To align with this convention when sampling filters, we convert from our original block covariance matrix representation to the representation above by simple reassignment of matrix entries, given by

$$\Sigma'_{ki+j,k\ell+m} := [\Sigma_{i,j}]_{\ell,m} \text{ for } 1 \leq i, j, \ell, m \leq k. \tag{1}$$

In this form, we may now easily generate a filter $F \in \mathbb{R}^{k \times k}$ by drawing a sample $f \in \mathbb{R}^{k^2}$ from $\mathcal{N}(\mu, \Sigma')$ and assigning $F_{i,j} := f_{ki+j}$. In this paper, we assume covariance matrices are in the block form unless we are sampling from a distribution, where the conversion between forms is assumed.

**Scope** We restricted our study to the large-filter depthwise convolutions found in new ViT-style CNNs, namely the popular ConvMixer and ConvNeXt architectures. These networks consist of a patch embedding layer followed by alternating spatial- and channel-mixing steps. Both use depthwise convolution for spatial mixing, but ConvMixer uses pointwise convolution (equivalently, linear layers) for spatial mixing while ConvNeXt uses MLPs. ConvMixer uses no internal downsampling, while ConvNeXt includes several downsampling stages. Unlike normal convolutions, the filters in depthwise convolutions act on each input channel separately rather than summing features over input channels. The *depth* of networks throughout the paper is synonymous with the number of depthwise convolutional layers. All networks investigated use a fixed filter size throughout the network, though the methods we present could easily be extended to the non-uniform case. Further, all methods presented do not concern the biases of convolutional layers.

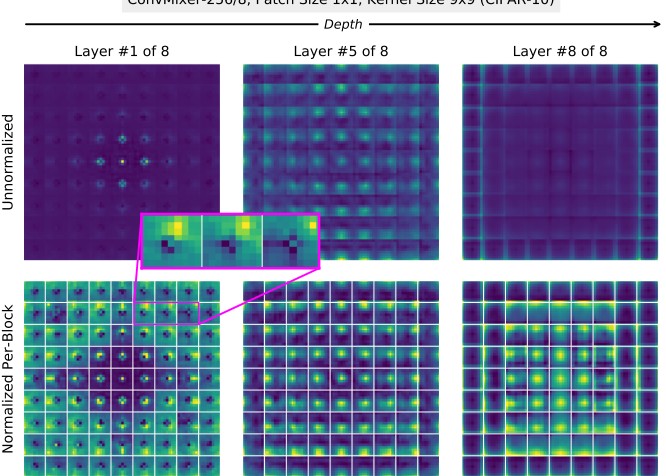

Figure 1: In pre-trained models, the covariance matrices of convolutional filters are *highly-structured*. Filters in earlier layers tend to be focused, becoming more diffuse as depth increases. Observing the structure of each block, we note that there is often a static, centered negative component and a dynamic positive component that moves according to the block's position. Often, covariances are higher towards the center of the filters.

## 2 THE COVARIANCES OF TRAINED CONVOLUTIONAL FILTERS AND THEIR TRANSFERABILITY ACROSS ARCHITECTURES

In this section, we propose a simple starting point in our investigation of convolutional filter covariance structure: using the distribution of filters from *pre-trained models* to initialize filters in new models, a process we term *covariance transfer*. In the simplest case, we use a pre-trained model with exactly the same architecture as the model to be initialized; we then show that we can actually transfer filter covariances across very different models.

**Basic method.** We use $i \in 1, \ldots, D$ to denote the $i^{\text{th}}$ depthwise convolutional layer of a model with $D$ layers. We denote the $j \in 1, \ldots, H$ filters of the $i^{\text{th}}$ pre-trained layer of the model by $F_{ij}$ for a model with $H$ convolutional filters in a particular layer (*i.e.,* hidden dimension $H$) and $F'$ to denote the filters of a new, untrained model. Then the empirical covariance of the filters in layer $i$ is

$$\Sigma_i = \text{Cov}[\text{vec}(F_{i1}), \ldots, \text{vec}(F_{iH})], \quad (2)$$

with the mean $\mu_i$ computed similarly. Then the new model can be initialized by drawing filters from the multivariate Gaussian distribution with parameters $\mu_i, \Sigma_i$:

$$F'_{ij} \sim \mathcal{N}(\mu_i, \Sigma_i) \quad \text{for } j \in 1, \ldots, H, i \in 1, \ldots, D \quad (3)$$

Note that in this section, we use the means of the filters in addition to the covariances to define the distributions from which to initialize. However, we found that the mean can be assumed to be zero with little change in performance, and we focus solely on the covariance in later sections.

**Experiment design.** We test our initialization methods primarily on ConvMixer since it is simple and exceptionally easy to train on CIFAR-10. We use FFCV (Leclerc et al., 2022) for fast data loading using our own implementations of fast depthwise convolution and RandAugment (Cubuk et al., 2020). To demonstrate the performance of our methods across a variety of training times, we train for 20, 50, or 200 epochs with a batch size of 512, and we repeat all experiments with three random seeds. For all experiments, we use a simple triangular learning rate schedule (see Appendix A.1) with the AdamW optimizer, a learning rate of .01, and weight decay of .01 as in Trockman & Kolter (2022).

Most of our CIFAR experiments use a ConvMixer-256/8 with either patch size 1 or 2; a ConvMixer-$H/D$ has precisely $D$ depthwise convolutional layers with $H$ filters each, ideal for testing our initial covariance transfer techniques. We train ConvMixers using popular filter sizes 3, 7, and 9, as well as 15. We also test our methods on ConvNeXt (Liu et al., 2022b), which includes downsampling unlike ConvMixer; we use a patch size of 1 or 2 with ConvNeXt rather than the default 4 to accomodate relatively small CIFAR-10 images, and the default $7 \times 7$ filters.

For most experiments, we provide two baselines for comparison: standard uniform initialization, the standard in PyTorch (He et al., 2015), as well as *directly* transferring the learned filters from a pre-trained model to the new model. In most cases, we expect new random initializations to fall between the performance of uniform and direct transfer initializations. For our covariance transfer experiments, we trained a variety of reference models from which to compute covariances; these are all trained for the full 200 epochs using the same settings as above.

**Frozen filters.** Cazenavette et al. noticed that ConvMixers with $3 \times 3$ filters perform well even when the filters are *frozen*; that is, the filter weights remain un-changed over the course of training, receiving no gradient updates. As we are initializing filters from the distribution of trained filters, we suspect that additional training may not be completely neces-sary. Consequently, in all experiments we investigate both models with *thawed* filters as well as their *frozen* counterparts. Freezing filters removes one of the two gradient calculations from depthwise convolution, resulting in substantial training speedups as kernel size increases (see Figure 2). ConvMixer-512/12 with kernel size $9 \times 9$ is around 20% faster, while $15 \times 15$ is around 40% faster. Fur-ther, good performance in the frozen filter setting suggests that an initialization technique is highly effective.

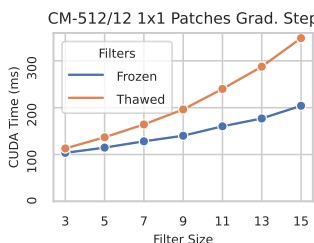

Figure 2: The backward pass is faster with frozen filters.

## 2.1 RESULTS

The simplest case of covariance transfer (from exactly the same architecture) is a fairly effective ini-tialization scheme for convolutional filters. In Fig. 3, note that this case of covariance transfer (group **B**) results in somewhat higher accuracies than uniform initialization (group **A**), particularly for 20-epoch training; it also substantially improves the case for frozen filters. Across all trials, the effect of using this initialization is higher for larger kernel sizes. In Fig. 8, we show that covariance transfer (*gold*) initially increases convergence, but the advantange over uniform initialization quickly fades. As expected, covariance transfer tends to fall between the performance of *direct transfer*, where we directly initialize using the filters of the pre-trained model, and default uniform initialization (see group **D** in Fig. 3 and the *green* curves in Fig. 8).

However, we acknowledge that it is not appealing to pre-train models just for an initialization tech-nique with rather marginal gains, so we explore the feasibility of covariance transfer from *smaller* models, both in terms of width and depth.

**Narrower models.** We first see if it's possible to train a narrower reference model to calculate filter covariances to initialize a wider model; for example, using a ConvMixer-32/8 to initialize a ConvMixer-256/8. In Figure 4, we show that the optimal performance surprisingly *comes from the covariances of a smaller model*. For filter sizes sizes greater than 3, the covariance transfer performance increases with width until width 32, and then decreases for width 256 for both the thawed and frozen cases. We plot this method in Fig. 3 (group **C**), and note that it almost uniformly exceeds the performance of covariance transfer from the same-sized model. Note that the method does not change; the covariances are simply calculated from a smaller sample of filters.

**Shallower models.** Covariance transfer from a shallow model to a deeper model is somewhat more complicated, as there is no longer a one-to-one mapping between layers. Instead, we *linearly interpolate* the covariance matrices to the desired depth (see Appendix A.1 for more details). Surprisingly, we find that this technique is also highly effective: for example, for a 32-layer-deep ConvMixer, the optimal covariance transfer result is from an 8-layer-deep ConvMixer, and 4-deep models are also quite effective (see Figure 4).

**Different patch sizes.** Similarly, it is straightforward to transfer covariances between models with different patch sizes. We find that initializing ConvMixers with $1 \times 1$ patches from filter covariances of ConvMixers with $2 \times 2$ patches leads to a decrease in performance relative to using a reference model of the correct patch size; however, using the filters of a $1 \times 1$ patch-size ConvMixer to initialize a $2 \times 2$ patch size ConvMixer increases performance (see group **b** vs. group **B** in Fig. 9). Yet, in both cases, the performance is better than uniform initialization.

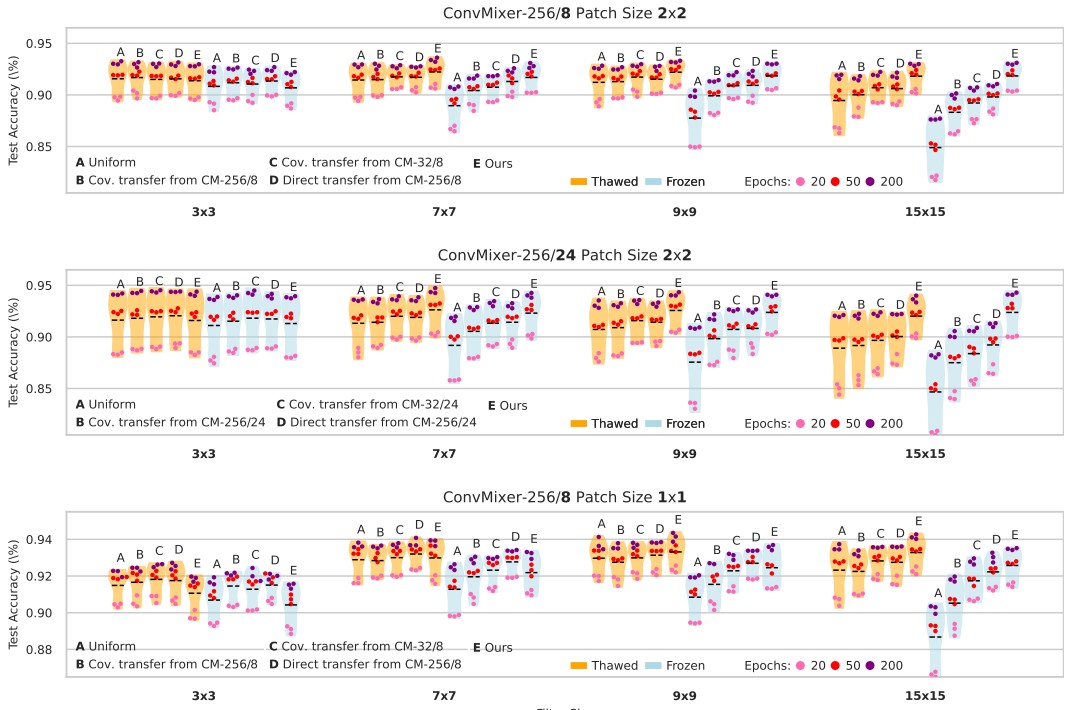

Figure 3: CIFAR-10 accuracy for uniform initialization (**A**), baseline covariance transfer (**B-D**), and our custom initialization results (**E**).

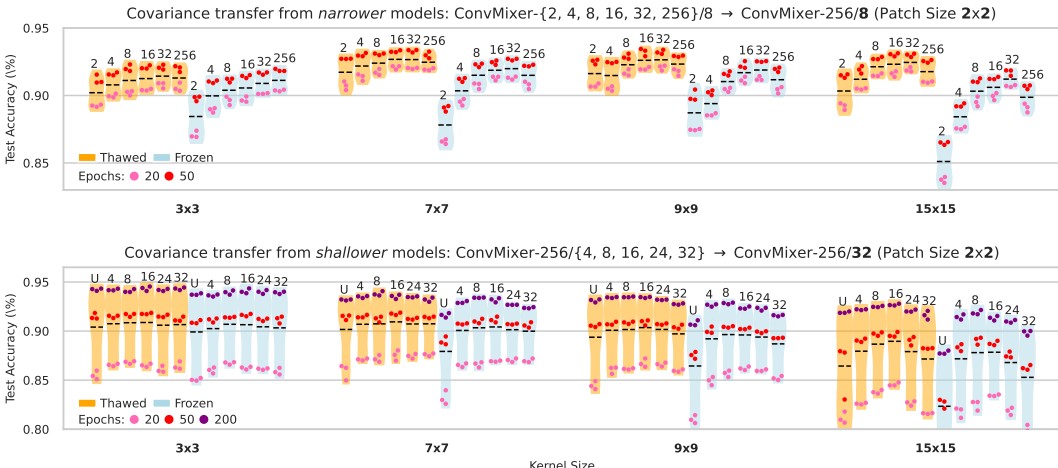

Figure 4: CIFAR-10 experimental results from initializing via covariances from narrower *(top)* and shallower *(bottom)* models. The numeric annotations represent the width *(top)* and depth *(bottom)* of the pre-trained model we use to initialize. **U** represents uniform initialization.

**Different filter sizes.** Covariance transfer between models with different filter sizes is more challenging, as the covariance matrices have different sizes. In the block form, we mean-pad or clip each block to the target filter size, and then bilinearly interpolate over the blocks to reach a correctly-sized covariance matrix. This technique is still better than uniform initialization for filter sizes larger than 3 (which naturally has very little structure to transfer), especially in the frozen case (see Fig. 9)

**Discussion.** We have demonstrated that it is possible to initialize filters from the covariances of pre-trained models of different widths, depths, patch sizes, and kernel sizes; while some of these techniques perform better than others, they are almost all better than uniform initialization. Our ob-

servations indicate that the optimal choice of reference model is narrower or shallower, and perhaps with a smaller patch size or kernel size. We also found that covariance transfer from ConvMixers trained on ImageNet led to greater performance still (Appendix A). This suggests that the best covariances for filter initialization may be quite unrelated to the target model, *i.e.,* model independent.

## 3   D.I.Y. FILTER COVARIANCES

Ultimately, the above methods for initializing convolutional filters via transfer are limited by the necessity of a trained network from which to form a filter distribution, which must be accessible at initialization. We thus use observations on the structure of filter covariance matrices to construct our own covariance matrices from scratch. Using our construction, we propose a depth-dependent but simple initialization strategy for convolutional filters that greatly outperforms previous techniques.

**Visual observations.**   Filter covariance matrices in pre-trained ConvMixers and ConvNeXts have a great deal of structure, which we observe across models with different patch sizes, architectures, and data sets; see Fig. 1 and 32 for examples. In both the block and rearranged forms of the covariance matrices, we noticed clear repetitive structure, which led to an initial investigation on modeling covariances via Kronecker factorizations; see Appendix A for experimental results. Beyond this, we first note that the overall variance of filters tends to increase with depth, until breaking down towards the last layer. Second, we note that the blocks of the covariances often have a *static* negative component in the center, with a *dynamic* positive component whose position mirrors that of the block itself. Finally, the covariance of filter parameters is greater in their center, *i.e.,* covariance matrices are at first centrally-focused and become more diffuse with depth. These observations agree with intuition about the structure of convolutional filters: most filters have the greatest weight towards their center, and their parameters are correlated with their neighbors.

**Constructing covariances.**   With these observations in mind, we propose a construction of covariance matrices. We fix the (odd) filter size $k \in \mathbb{N}^+$, let $\mathbf{1} \in \mathbb{R}^{k \times k}$ be the all-ones matrix, and, as a building block for our initialization, use unnormalized Gaussian-like filters $Z_\sigma \in \mathbb{R}^{k \times k}$ with a single variance parameter $\sigma$, defined elementwise by

$$(Z_\sigma)_{i,j} := \exp\left( -\frac{(i - \lceil \frac{k}{2} \rceil)^2 + (j - \lceil \frac{k}{2} \rceil)^2}{2\sigma} \right) \text{ for } 1 \leq i, j, \leq k. \tag{4}$$

Such a construction produces filters similar to those observed in the blocks of the Layer #5 covariance matrix in Fig. 1.

To capture the *dynamic* component that moves according to the position of its block, we define the block matrix $C \in \mathbb{R}^{k^2 \times k^2}$ with $k \times k$ blocks by

$$[C_{i,j}] = \text{Shift}(Z_\sigma, i, j) \tag{5}$$

where the Shift operation translates each element of the matrix $i$ and $j$ positions forward in their respective dimensions; see Appendix E for details. We then define two additional components, both constructed from Gaussian filters: a *static* component $S = \mathbf{1} \otimes Z_\sigma \in \mathbb{R}^{k^2 \times k^2}$ and a blockwise mask component $M = Z_\sigma \otimes \mathbf{1} \in \mathbb{R}^{k^2 \times k^2}$, which encodes higher variance as pixels approach the center of the filter.

Using these components and our intuition, we first consider $\hat{\Sigma} = M \odot (C - \frac{1}{2}S)$, where $\odot$ is an elementwise product. While this adequately represents what we view to be the important structural components of filter covariance matrices, it does not satisfy the property $[\Sigma_{i,j}]_{\ell,m} = [\Sigma_{\ell,m}]_{i,j}$ (*i.e.,* covariance matrices must be symmetric, accounting for our block representation). Consequently, we instead calculate its symmetric part, using the notation as follows to denote a "block-transpose":

$$\Sigma^B = \Sigma' \iff [\Sigma_{i,j}]_{\ell,m} = [\Sigma'_{\ell,m}]_{i,j} \text{ for } 1 \leq i, j, \ell, m \leq k. \tag{6}$$

Equivalently, this is the perfect shuffle permutation such that $(X \otimes Y)^B = Y \otimes X$ with $X, Y \in \mathbb{R}^{k \times k}$. First, we note that $C^B = C$ due to the definition of the shift operation used in Eq. 5 (see Appendix E). Then, noting that $S^B = M$ and $M^B = S$ by the previous rule, we define our construction

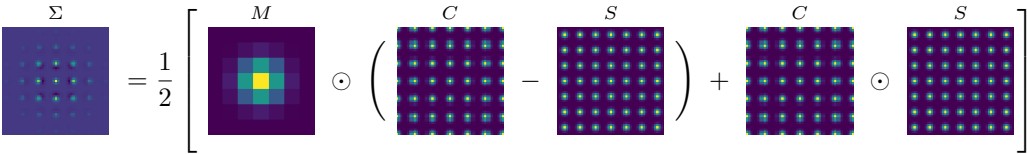

Figure 5: Our convolutional covariance matrix construction with $\sigma = \pi/2$.

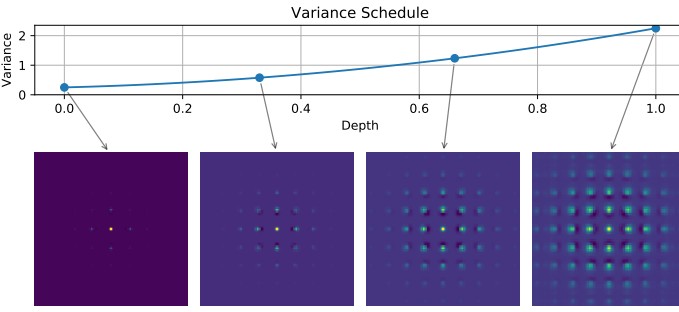

Figure 6: How our initialization changes with depth. Variance increases quadratically with depth according to a schedule which can be chosen through visual inspection of pre-trained models or through grid search. Here we use the parameters $\sigma_0 = .5$, $v_\sigma = .5$, $a_\sigma = 3$.

of $\Sigma$ to be the symmetric part of $\hat{\Sigma}$ (where $C, S, M$ are implicitly parameterized by the $\sigma$ of $Z_\sigma$):

$$\Sigma = \tfrac{1}{2}(\hat{\Sigma} + \hat{\Sigma}^T) = \tfrac{1}{2}\left[M \odot (C - \tfrac{1}{2}S) + (M \odot (C - \tfrac{1}{2}S))^B\right] \tag{7}$$

$$= \tfrac{1}{2}\left[M \odot (C - \tfrac{1}{2}S) + (M^B \odot (C^B - \tfrac{1}{2}S^B))\right] = M \odot (C - \tfrac{1}{2}S) + S \odot (C - \tfrac{1}{2}M) \tag{8}$$

$$= \tfrac{1}{2}\left[M \odot (C - S) + S \odot C\right]. \tag{9}$$

While $\Sigma$ is now symmetric (in the rearranged form of Eq. 1), it is not positive semi-definite, but can easily be projected to $\mathbb{S}_+^{k^2}$, as is often done automatically by multivariate Gaussian procedures. We illustrate our construction in Fig. 5, and provide an implementation in Fig. 30.

**Completing the initialization.** As explained in Fig. 1, we observed that in pre-trained models, the filters become more "diffuse" as depth increases; we capture this fact in our construction by increasing the parameter $\sigma$ with depth according to a simple quadratic schedule; let $d$ be the percentage depth, i.e., $d = \frac{i-1}{D-1}$ for the $i^{\text{th}}$ convolutional layer of a model with $D$ total such layers. Then for layer $i$, we parameterize our covariance construction by a *variance schedule*:

$$\sigma(d) = \sigma_0 + v_\sigma d + \tfrac{1}{2}a_\sigma d^2 \tag{10}$$

where $\sigma_0, v_\sigma, a_\sigma$ jointly describe how the covariance evolves with depth. Then, for each layer $i \in 1, \ldots, D$, we compute $d = \frac{i-1}{D-1}$ and initialize the filters as $F_{i,j} \sim \mathcal{N}(0, \Sigma'_{\sigma(d)})$ for $j \in 1, \ldots, H$. We illustrate our complete initialization scheme in Figure 6.

## 4 RESULTS

In this section, we present the performance of our initialization within ConvMixer and ConvNeXt on CIFAR-10 and ImageNet classification, finding it to be highly effective, particularly for deep models with large filters. Our new initialization overshadows our previous covariance transfer results.

Settings of initialization hyperparameters $\sigma_0$, $v_\sigma$, and $a_\sigma$ were found and fixed for CIFAR-10 experiments, while two such settings were used for ImageNet experiments. Appendix C.1 contains full details on our (relatively small) hyperparameter searches and experimental setups, as well as empirical evidence that our method is *robust to a large swath of hyperparameter settings*. Additional experiments on more datasets and baseline initializations may be found in Appendix B.

### 4.1 CIFAR-10 RESULTS

**Thawed filters.** In Fig. 3, we show that large-kernel models using our initialization (group **E**) outperform those using uniform initialization (group **A**), covariance transfer (groups **B, C**), and

even those directly initializing via learned filters (group **D**). For $2 \times 2$-patch models (200 epochs), relative to uniform, our initialization causes up to a 1.1% increase in accuracy for ConvMixer-256/8, and up to 1.6% for ConvMixer-256/24. The effect size increases with the the filter size, and is often more prominent for shorter training times. Results are similar for $1 \times 1$-patch models, but with a smaller increase for $7 \times 7$ filters (0.15% vs. 0.5%). Our initialization has the same effects for ConvNeXt (Fig. 7). However, our method works poorly for $3 \times 3$ filters, which we believe have fundamentally different structure than larger filters; this setting is better-served by our original covariance transfer techniques.

In addition to improving the final accuracy, our initialization also drastically speeds up convergence of models with thawed filters (see Fig. 8), particularly for deeper models. A ConvMixer-256/16 with $2 \times 2$ patches using our initialization reaches 90% accuracy in approximately 50% fewer epochs than uniform initialization, and around 25% fewer than direct learned filter transfer. The same occurs, albeit to a lesser extent, for $1 \times 1$ patches—but note that for this experiment we used the same initialization parameters for both patch sizes to demonstrate robustness to parameter choices.

**Frozen filters.** Our initialization leads to even more surprising effects in models with frozen filters. In Fig. 3, we see that frozen-filter $2 \times 2$-patch models using our initialization often *exceed the performance of their uniform, thawed-filter counterparts* by a significant margin of 0.4% – 2.0% for 200 epochs, and an even larger margin of 0.6% – 5.0% for 20 epochs (for large filters). That is, group **E** *(frozen)* consistently outperforms groups **A-D** *(thawed)*, and in

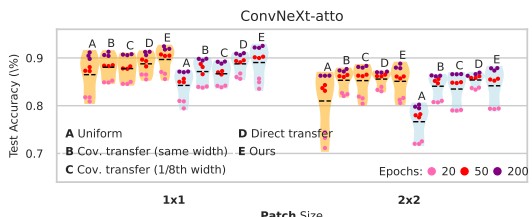

Figure 7: Our init also improves ConvNeXt's accuracy on CIFAR-10 (group **E** *vs.* **A**).

some cases even group **E** *(thawed)*, especially for the deeper 24-layer ConvMixer. While this effect breaks down for $1 \times 1$ patch models, such frozen-filter models still see accuracy increases of 0.6%–3.5%. However, the effect can still be seen for $1 \times 1$-patch ConvNeXts (Fig. 7). Also note that frozen-filter models can be up to 40% faster to train (see Fig. 2), and may be more robust (Cazenavette et al.).

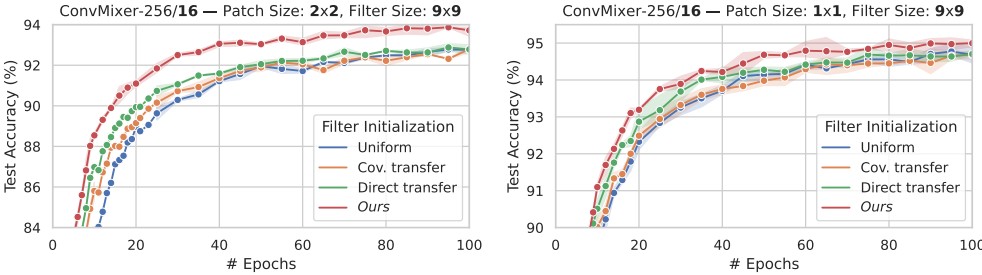

Figure 8: Convergence plots: each data point runs through a full cycle of the LR schedule, and all points are averaged over three trials with shaded standard deviation.

## 4.2 IMAGENET EXPERIMENTS

Our initialization performs extremely well on CIFAR-10 for large-kernel models, almost always helping and rarely hurting. Here, we explore if the performance gains transfer to larger-scale ImageNet models. We observe in Fig. 32, Appendix F that filter covariances for such models have finer-grained structure than models trained on CIFAR-10, perhaps due to using larger patches. Nonetheless, our initialization leads to quite encouraging improvements in this setting.

**Experiment design.** We used the "A1" training recipe from Wightman et al. (2021), with cross-entropy loss, fewer epochs, and a triangular LR schedule as in Trockman & Kolter (2022). We primarily demonstrate our initialization for 50-epoch training, as the difference between initializations is most pronounced for lower training times. We also present two full, practical-scale 150-epoch experiments on large models. We also included covariance transfer experiments in Appendix F.

Table 1: ImageNet-1k accuracy from various architectures and initializations. "Ours" denotes our proposed initialization. **Bold** indicates best within architecture and category (frozen or thawed).

| Model | | | | THAWED | | | FROZEN | | |
|---|---|---|---|---|---|---|---|---|---|
| Architecture | Filter Size | Patch Size | # Epochs | Uniform | Ours .15 .5 .25 | Ours .15 .25 1.0 | Uniform | Ours .15 .5 .25 | Ours .15 .25 1.0 |
| ConvMixer-512/12 | 9 | 14 | 50 | 67.03 | **67.41** | 67.34 | 60.47 | **64.43** | 64.12 |
| ConvMixer-512/24 | 9 | 14 | 50 | 67.76 | **69.60** | 69.52 | 62.50 | **66.57** | 66.38 |
| ConvMixer-512/32 | 9 | 14 | 50 | 65.00 | 68.78 | **68.84** | 55.79 | **66.59** | 66.32 |
| ConvMixer-1024/12 | 9 | 14 | 50 | 73.55 | 73.62 | **73.75** | 68.96 | **71.48** | 71.30 |
| ConvMixer-1024/24 | 9 | 14 | 50 | 74.19 | 75.33 | **75.50** | 69.65 | **73.42** | 74.31 |
| ConvMixer-1024/32 | 9 | 14 | 50 | 72.18 | **74.98** | 74.95 | 64.94 | 73.00 | **73.12** |
| ConvMixer-512/12 | 9 | 7 | 50 | 72.05 | 71.92 | **72.32** | 67.25 | 68.91 | **68.92** |
| ConvNeXt-Atto | 7 | 4 | 50 | **69.96** | 67.84 | 68.06 | 51.43 | **64.52** | 64.43 |
| ConvNeXt-Tiny | 7 | 4 | 50 | 75.99 | 76.08 | **77.11** | 64.17 | 74.62 | **75.21** |
| ConvMixer-1536/24 | 9 | 14 | 150 | 80.11 | | **80.28** | | | |
| ConvNeXt-Tiny | 7 | 4 | 150 | 79.74 | | **79.81** | | | |

**Thawed filters.** On models trained for 50 epochs with thawed filters, our initialization improves the final accuracy by $0.4\% - 3.8\%$ (see Table 1). For the relatively-shallow ConvMixer-512/12 on which we tuned the initialization parameters, we see a gain of just $0.4\%$; however, when increasing the depth to 24 or 32, we see larger gains of $1.8\%$ and $3.8\%$, respectively, and a similar trend among the wider ConvMixer-1024 models. Our initialization also boosts the accuracy of the 18-layer ConvNeXt-Tiny from $76.0\%$ to $77.1\%$; however, it decreased the accuracy of the smaller, 12-layer ConvNeXt-Atto. This is perhaps unsurprising, seeing as our initialization seems to be more helpful for deep models, and we used hyperparameters optimized for a model with a substantially different patch and filter size.

Our initialization is also beneficial for more-practical 150-epoch training, boosting accuracy by around $0.1\%$ on both ConvMixer-1536/24 and ConvNeXt-Tiny (see Table 1, bottom rows). While the effect is small, this demonstrates that our initialization is still helpful even for longer training times and very wide models. We expect that within deeper models and with slightly more parameter tuning, our initialization could lead to still larger gains in full-scale ImageNet training.

**Frozen filters.** Our initialization is extremely helpful for models with frozen filters. Using our initialization, the difference between thawed and frozen-filter models decreases with increasing depth, *i.e.,* it leads to $2 - 11\%$ improvements over models with frozen, uniformly-initialized filters. For ConvMixer-1024/32, the accuracy improves from $64.9\%$ to $73.1\%$, which is over $1\%$ *better than the corresponding thawed, uniformly-initialized model*, and only $2\%$ from the best result using our initialization. This mirrors the effects we saw for deeper models on our earlier CIFAR-10 experiments. We see a similar effect for ConvNeXt-Tiny, with the frozen version using our initialization achieving $75.2\%$ accuracy *vs.* the thawed $76.0\%$. In other words, our initialization so effectively captures the structure of convolutional filters that it is hardly necessary to train them after initialization; one benefit of this is that it substantially speeds up training for large-filter convolutions.

## 5 CONCLUSION

In this paper, we proposed a simple, closed-form, and learning-free initialization scheme for large depthwise convolutional filters. Models using our initialization typically reach higher accuracies more quickly than uniformly-initialized models. We also demonstrated that our random initialization of convolutional filters is so effective, that in many cases, networks perform nearly as well (or even better) if the resulting filters do not receive gradient updates during training. Moreover, like the standard uniform initializations generally used in neural networks, our technique merely samples from a particular statistical distribution, and it is thus almost completely computationally free. ***In summary,*** *our initialization technique for the increasingly-popular large-kernel depthwise convolution operation almost always helps, rarely hurts, and is also free.*

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

## A  ADDITIONAL CIFAR RESULTS

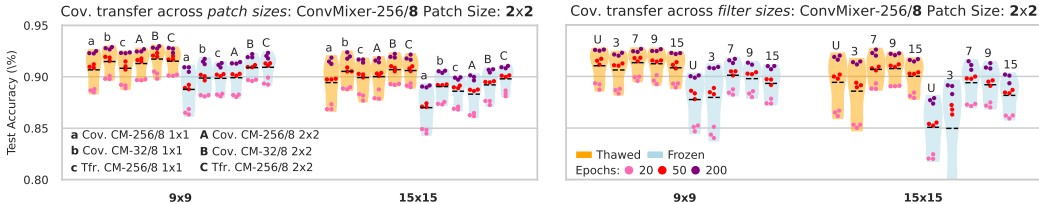

Figure 9: Initializing via covariances from models with different patch (*left*) and filter sizes (*right*). *Left:* Lowercase denotes initializing from patch size $1 \times 1$, and uppercase $2 \times 2$. *Right:* Annotations denote the reference filter size, **U** is uniform.

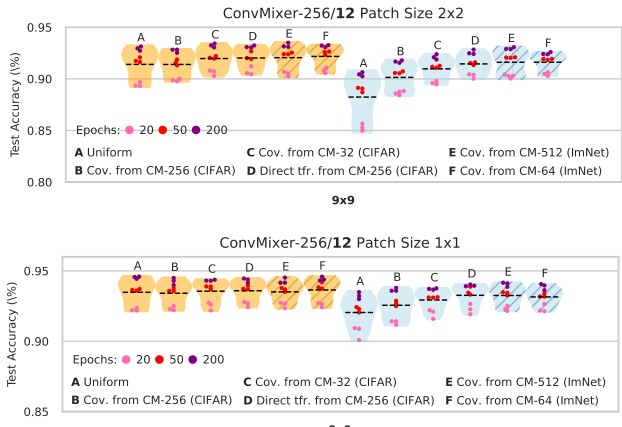

Figure 10: Using filter distributions from pre-trained ImageNet models to initialize models trained on CIFAR-10 is also effective (represented by groups **E** and **F**, with hatch marks).

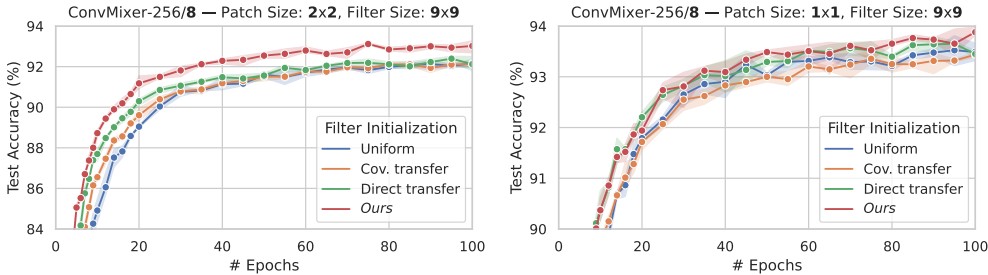

Figure 11: Convergence plots: each data point runs through a full cycle of the LR schedule, and all points are averaged over three trials with shaded standard deviation.

**Covariance structure.**  As a first step towards modeling the structure of filter covariances, we replaced covariances with their Kronecker-factorized counterparts using the rearranged form of the covariance matrix defined in Eq. (1), *i.e.,* $\Sigma = A \otimes A$ where $A \in \mathbb{S}_+^k$. Surprisingly, this slightly improved performance over unfactorized covariance transfer (see Fig. 12), suggesting that filter covariances are not only eminently transferrable for initialization, but that their core structure may be simpler than meets the eye. Kronecker factorizations were computed via gradient descent minimizing the mean squared error.

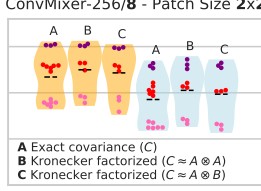

Figure 12: Approx. covs.

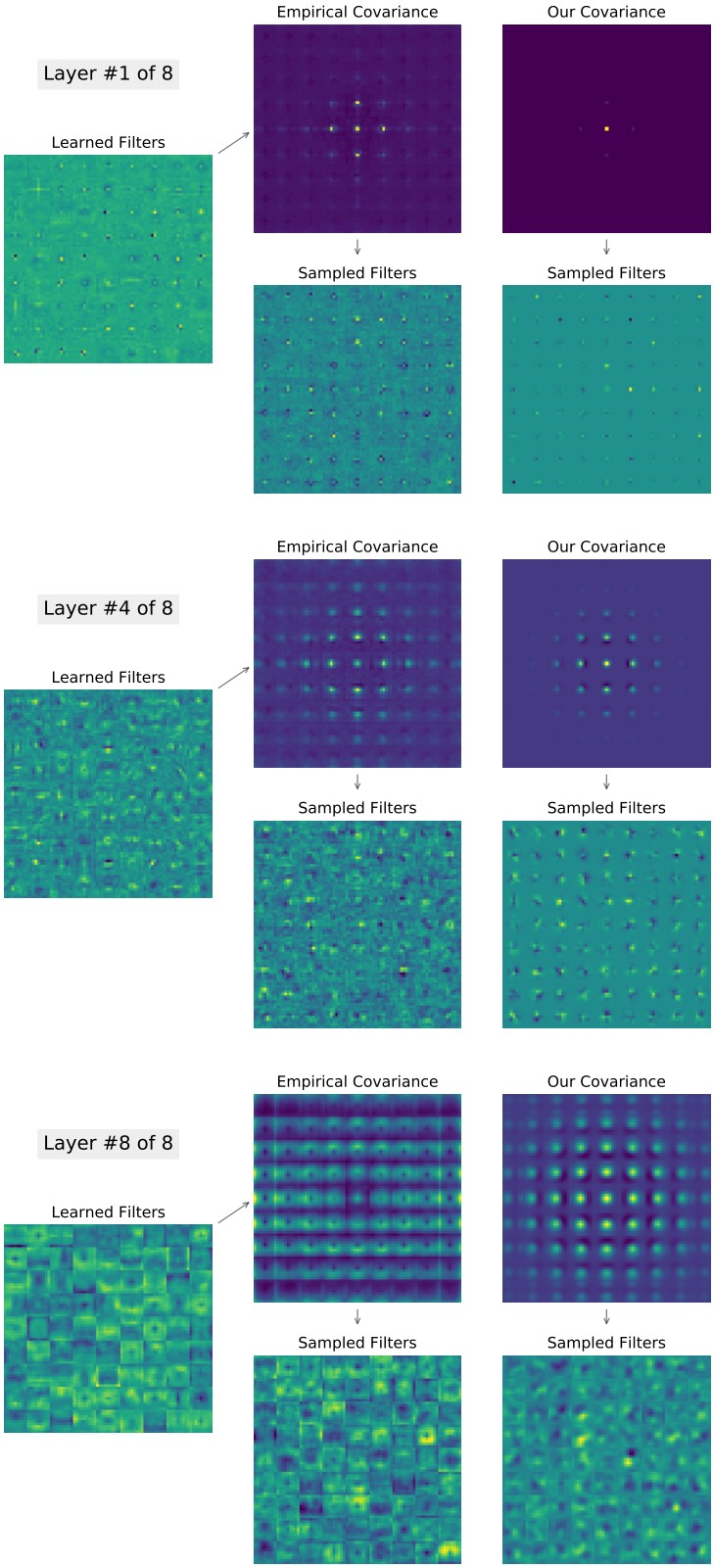

Figure 13: Filters learned or generated for ConvMixer-256/8 with $2 \times 2$ patches and $9 \times 9$ filters trained on CIFAR-10: learned filters (*left*), filters sampled from the Gaussian defined by the empirical covariance matrix of learned filters (*center*), and filters from our initialization technique (*right*).

## A.1 Additional Experiment Details

```
1   epochs = 100
2   lr_max = 0.01
3   lr_sched = lambda t: np.interp([t], \
4     [0, epochs*2//5, epochs*4//5, epochs], \
5     [0, lr_max, lr_max/20.0, 0])[0]
6   for epoch in range(epochs):
7     for i, (X, y) in enumerate(loader):
8       # ...
9       lr = lr_sched(epoch + (i + 1)/len(loader))
10      opt.param_groups[0].update(lr=lr)
11      # ...
```

Figure 14: Implementation and visualization of the piecewise triangular learning rate schedule we used for all experiments. We used the implementation from Trockman & Kolter (2022).

```
1   def linear_interpolation(covs, from_depth, to_depth):
2     # covs: list of covariance matrices (np.array)
3     # from_depth: depth of model from which we are transferring
4     # to_depth: depth of model we are transferring to
5     from_knots = np.arange(0, from_depth)/(from_depth - 1)
6     to_knots = np.arange(0, to_depth)/(to_depth - 1)
7     ret = []
8     for knot in to_knots:
9       for i in range(len(from_knots)-1):
10        if knot >= from_knots[i] and knot <= from_knots[i+1]:
11          a_weight = (knot - from_knots[i+1]) / (from_knots[i] - from_knots[i + 1])
12          b_weight = (knot - from_knots[i]) / (from_knots[i+1] - from_knots[i])
13          ret.append(a_weight * covs[i] + b_weight * covs[i+1])
14          break
15    return ret
```

Figure 15: Implementation of linear interpolation used for transferring covariances from shallower to deeper models. We treat the depth of a layer as a percent of the full depth of the network, *i.e.,* our notion of depth is always in the interval $[0, 1]$.

# B ADDITIONAL BASELINES AND DATASETS

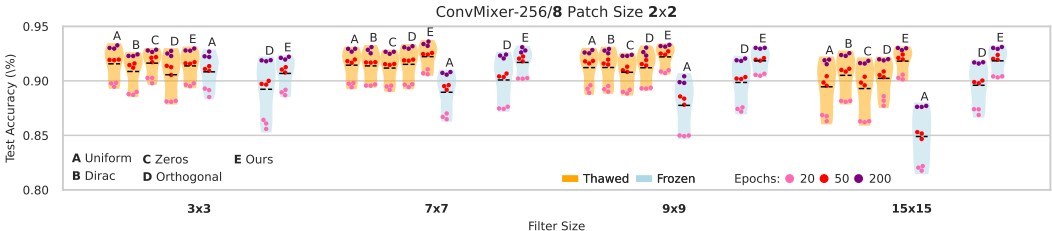

Figure 16: Our initialization technique outperforms alternatives (uniform, dirac, all-zero, and orthogonal initialization) for filter sizes larger than 3, especially for short-duration training. Note that for dirac and all-zero initialization (B, C) in the frozen case, accuracies are $\approx 60\%$ and thus are not on the graph. Results on CIFAR-10.

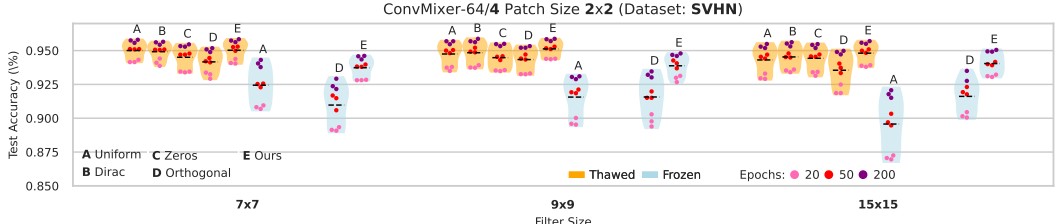

Figure 17: Our initialization also generally outperforms or matches baselines on the Street View House Numbers (SVHN) dataset. Since this task is relatively easy, we used a much smaller ConvMixer-64/4. The advantage of our method is most noticeable for short-duration (20 epoch) training and the frozen-filter setting.

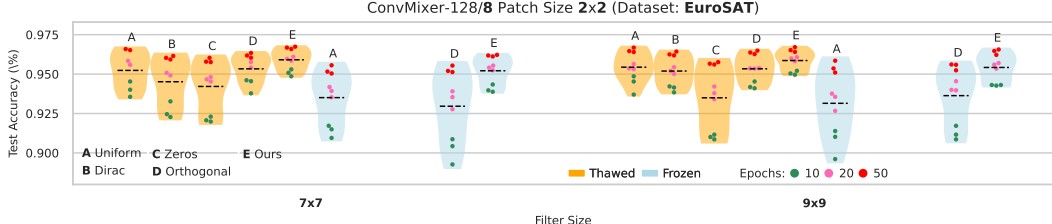

Figure 18: Our initialization is again competitive with baselines on the EuroSAT dataset, where we used a random 75/25 train/test split. We found our models converged quickly on this task, and thus used fewer epochs for all experiments (10, 20, and 50). Our initialization approximately matches uniform initialization and outperforms other methods in the 50-epoch setting, but performs better in the 10- and 20- epoch settings. It is also particularly effective for frozen filters.

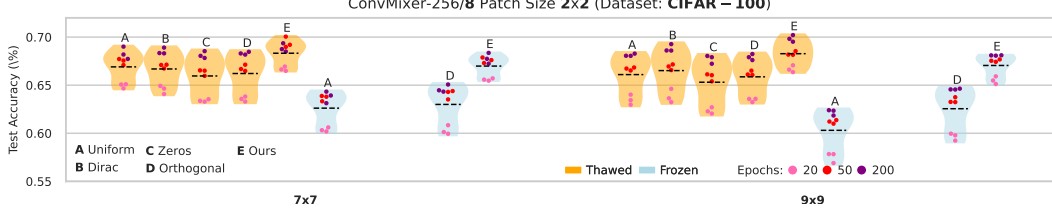

Figure 19: Our initialization is particularly advantageous on the more-difficult CIFAR-100 dataset, with 50-epoch training meeting or even surpassing 200-epoch uniformly-initialized training.

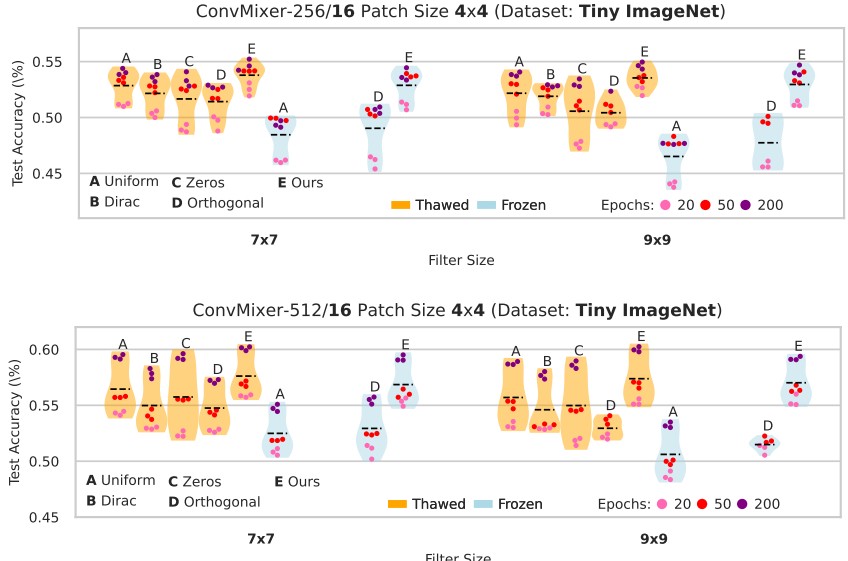

Figure 20: Our initialization is also helpful on Tiny ImageNet. Here we increased the patch size to 4x4 to accomodate the increased input size of 64x64 (for computational efficiency).

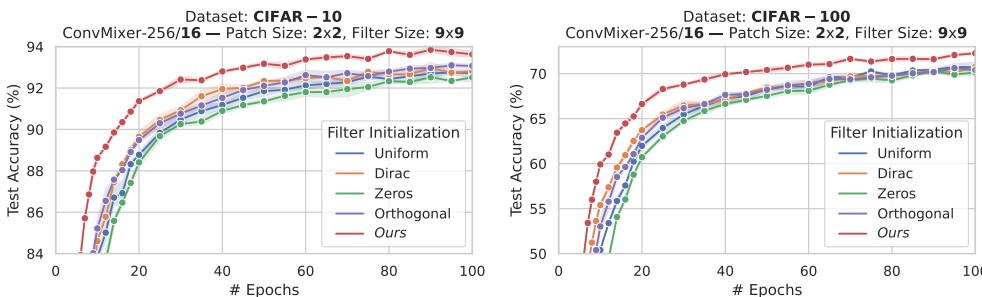

Figure 21: Compared to other initialization techniques, ours results in faster convergence when training a ConvMixer-256/16 on CIFAR-10 and CIFAR-100. That is, one can often train models for fewer epochs when using our initialization to achieve results comparable to those from using other initialization techniques.

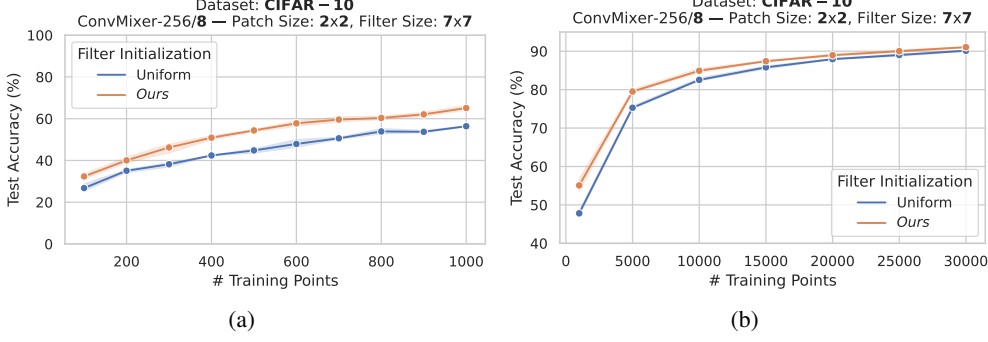

Figure 22: We investigated how our initialization affects data efficiency by training on random (smaller) subsets of CIFAR-10. Each trial is averaged over 3 such subsets. In the very low-data setting (a), we trained for 300 epochs to compensate for the smaller number of iterations overall. In (b), we trained for 100 epochs. In some cases, using our initialization is comparable to doubling the number of training points.

## C  HYPERPARAMETER GRID SEARCHES & EXPERIMENTAL SETUP

**CIFAR-10 hyperparameter search.**  We chose an initial setting of our method's three hyperparameters via visual inspection, and then refined them via small-scale grid searches. For CIFAR-10 experiments, we searched over parameters for ConvMixer-256/8 with frozen $9 \times 9$ filters trained for 20 epochs, and chose $\sigma_0 = .08, v_\sigma = .37, a_\sigma = 2.9$ for $2 \times 2$-patch models, and found the optimal parameters for $1 \times 1$-patch models to be approximately doubled. However, note that our initialization is quite robust to different parameter settings, with the difference from our doubling choice being less than $0.1\%$ (see Figure 23). We used the same parameters across all kernel sizes, as well as for ConvNeXt, a choice which is likely sub-optimal; our search only serves as a rough heuristic.

**ImageNet-1k hyperparameter search.**  We did a small grid search using a ConvMixer-512/12 with $14 \times 14$ patches and $9 \times 9$ filters trained for 10 epochs on ImageNet-1k (see Appendix F), from which we chose two candidate settings: $\sigma_0 = .15, v_\sigma = .5, a_\sigma = .25$ for frozen-filter models and $\sigma_0 = .15, v_\sigma = 0.25, a_\sigma = 1.0$ for thawed models. We use these parameters for all the ImageNet experiments, even for models with different patch and kernel sizes (*e.g.,* ConvNeXt). This demonstrates that *hyperparameter tuning is optional* for our technique; its transferability is not surprising given our results in Sec. 2.

### C.1  CIFAR-10 GRID SEARCHES

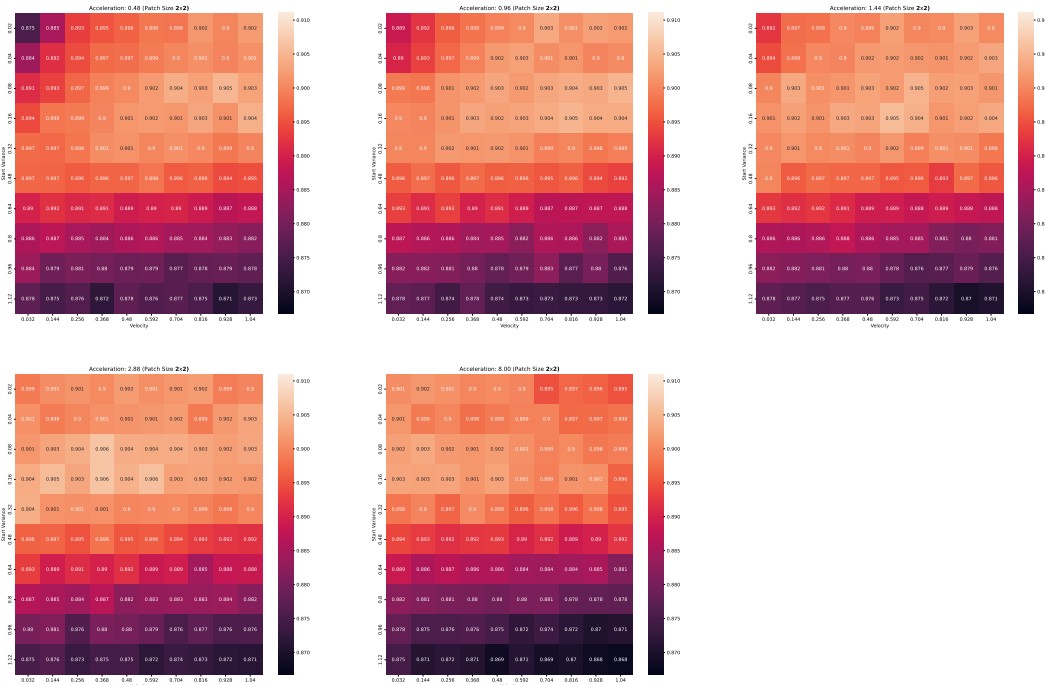

Figure 23: Grid search over initialization parameters $\sigma_0, v_\sigma, a_\sigma$ for ConvMixer-258/8 with $9 \times 9$ *frozen* filters and $2 \times 2$ patches trained for 20 epochs on CIFAR-10. Note that the performance of uniform initialization is only $\approx 85\%$, i.e., almost all choices result in *some* improvement.

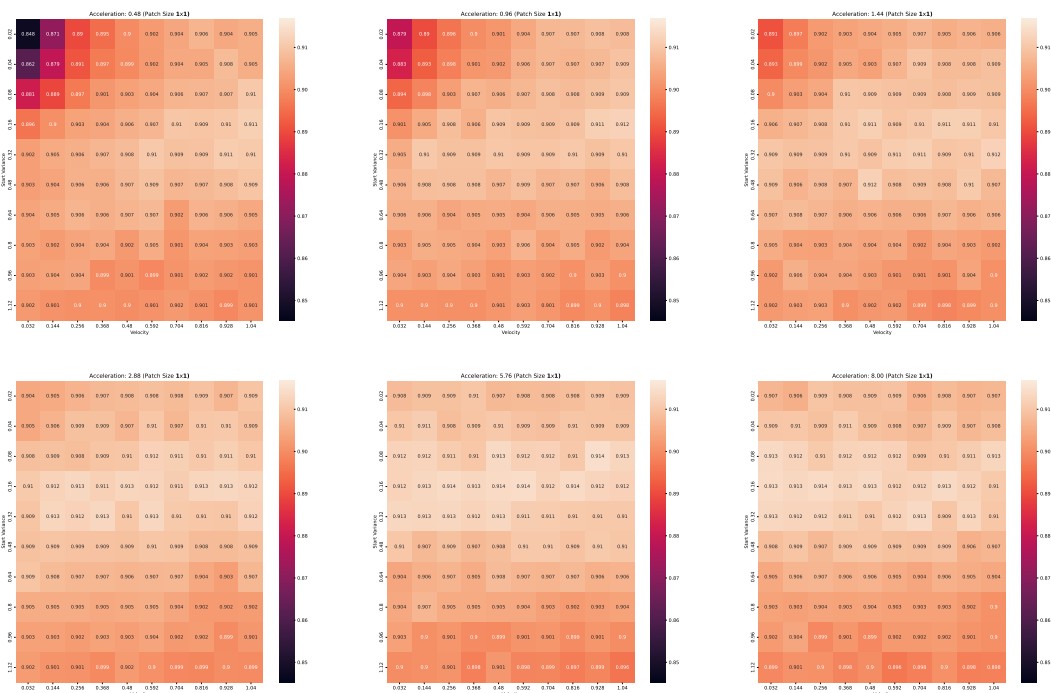

Figure 24: Grid search over initialization parameters $\sigma_0, v_\sigma, a_\sigma$ for ConvMixer-258/8 with $9 \times 9$ *frozen* filters and $1 \times 1$ patches trained for 20 epochs on CIFAR-10. Note that the performance of uniform initialization is only $\approx 88\%$, i.e., almost all choices result in *some* improvement.

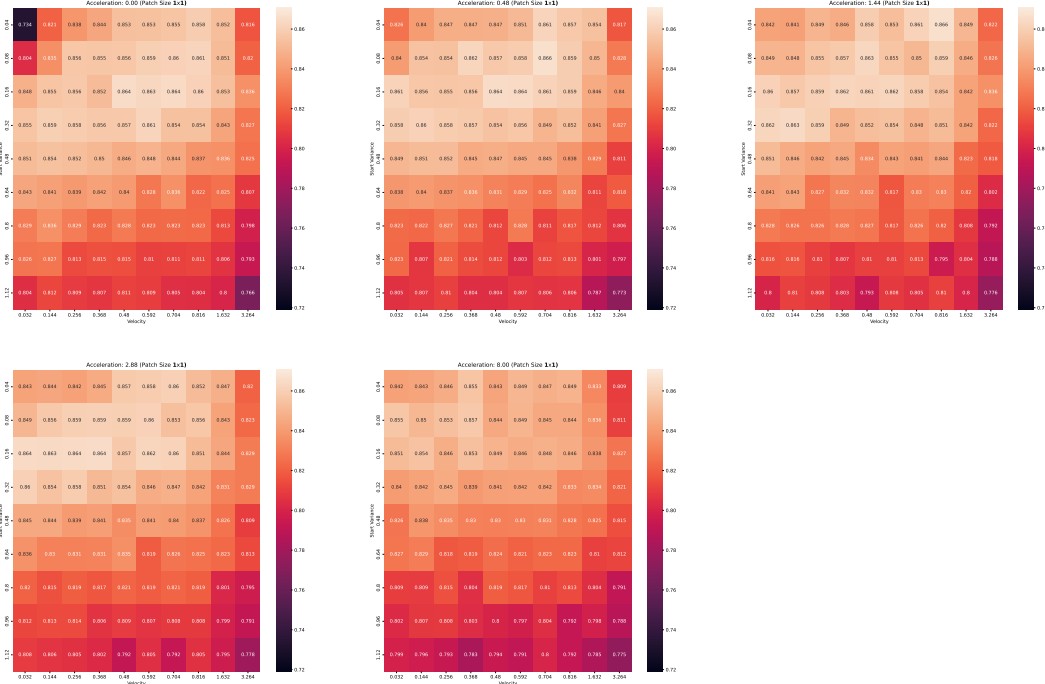

Figure 25: Grid search over initialiation parameters $\sigma_0, v_\sigma, a_\sigma$ for ConvNeXt-atto on CIFAR-10 with frozen filters and $1 \times 1$ patches trained for 20 epochs on CIFAR-10. Note the baseline performance with uniform initialization is around $80\%$, *i.e.,* compared to ConvMixer there are more potentially disadvantageous parameter combinations.

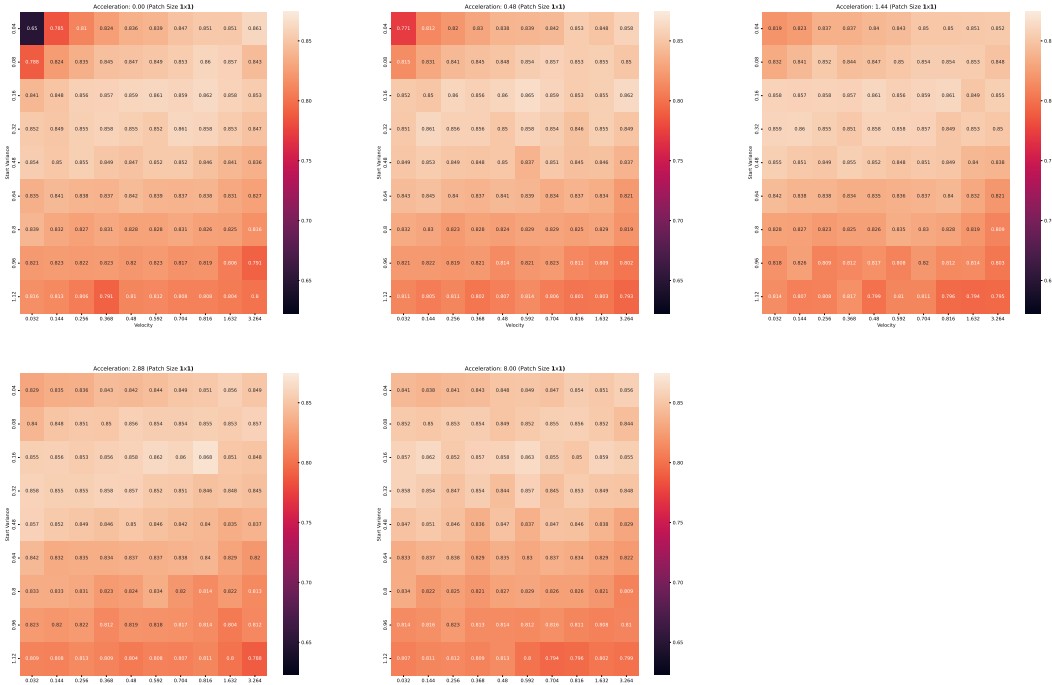

Figure 26: Grid search over initialiation parameters $\sigma_0, v_\sigma, a_\sigma$ for ConvNeXt-atto on CIFAR-10 with frozen filters and $1 \times 1$ patches trained for 20 epochs, using the "sawtooth" variance schedule (see Fig 27) to account for downsampling layers. While this perhaps shows better robustness to parameter changes than Fig. 26, the effect could also be due to effectively dividing the parameters by two.

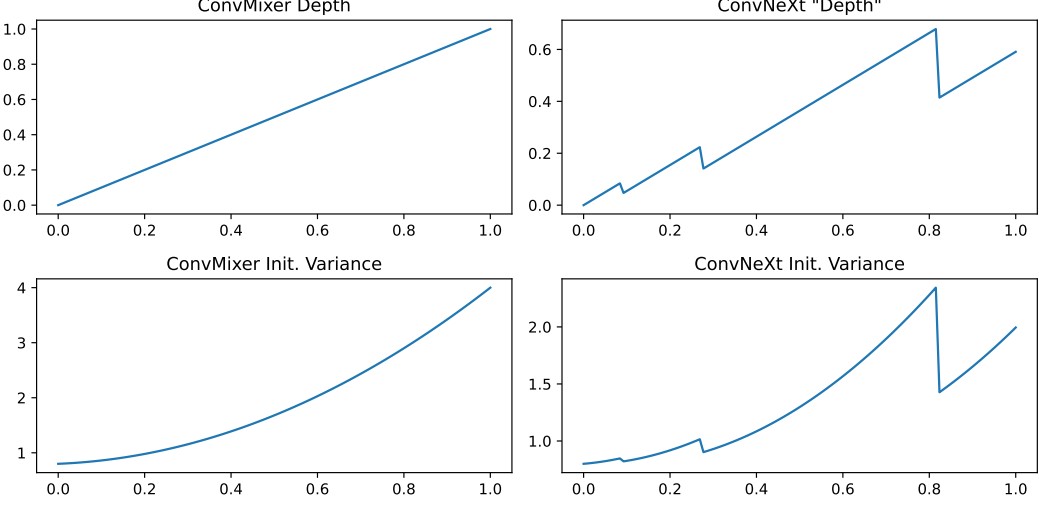

Figure 27: Proposed stepwise variance schedule for ConvNeXt, *i.e.,* a model including downsampling layers. In our experiments, we saw no advantage to using this scheme.

## C.2 IMAGENET GRID SEARCHES

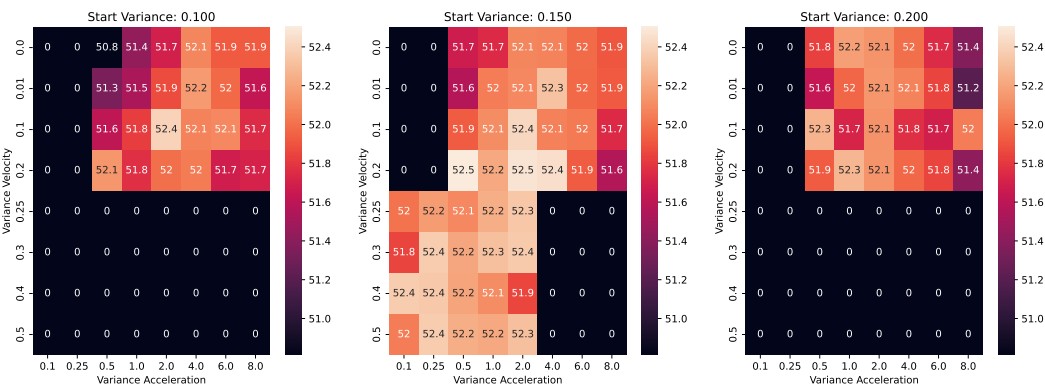

Figure 28: **Frozen filters:** Grid search over initialization parameters for ConvMixer-512/12 with $14 \times 14$ patches and $9 \times 9$ filters, 10 epochs.

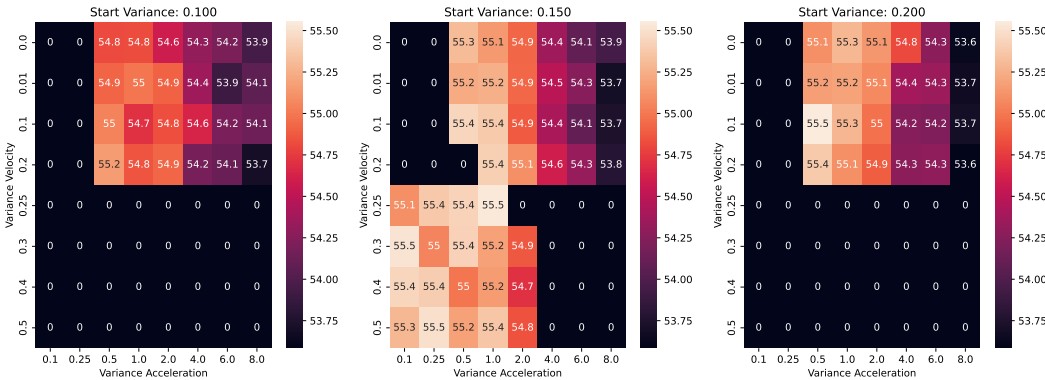

Figure 29: **Thawed filters:** Grid search over initialization parameters for ConvMixer-512/12 with $14 \times 14$ patches and $9 \times 9$ filters, 10 epochs.

# D IMPLEMENTATIONS

## D.1 TERSE IMPLEMENTATION

```
def ConvCov(k, s):                                def Gauss(k, mx, my, s):
  C = np.zeros((k**2,)*2)                           res = np.zeros((k, k))
  for i, j in np.ndindex(k,k):                      for i, j in np.ndindex(k,k):
    C[k*i:k*i+k,k*j:k*j+k] = Gauss(k,j,i,s)           cx,cy = (j-mx-k//2-1)%k,(i-my-k//2-1)%k
  Z,l = Gauss(k,k//2,k//2,s),np.ones((k,k))           z = ((cx-k//2)**2+(cy-k//2)**2)/s
  S, M = np.kron(l, Z),np.kron(Z, l)                  res[i, j] = np.exp(-0.5*z)
  return 0.5 * (M * (C - S) + C * S)                return res.reshape(k, k)
```

Figure 30: Implementation of our convolution covariance construction in NumPy.

```
def Initialize(wconv, d, s0, sv, sa):
  c, _, ks, _ = wconv.shape
  s = s0 + sv * d + 0.5 * sa * d**2
  cov = ConvCov(ks, s).reshape((ks,)*4).transpose(0,2,1,3).reshape((ks**2,)*2)
  filters = np.random.multivariate_normal(np.zeros(ks**2), cov, size=(c,))
  wconv.data = torch.tensor(filters.reshape(c,1,ks,ks),dtype=wconv.dtype,device=wconv.device)

# Find depthwise convolutional layers
convs = [x for x in model.modules() if isinstance(x, nn.Conv2d) \
  and len(x.weight.shape) == 4 and x.weight.shape[1] == 1]

# Initialize them according to variance schedule
for i, conv in enumerate(convs):
  Initialize(conv.weight, i / (len(convs) - 1), 0.16, 0.32, 2.88)
```

Figure 31: Code to use our covariance construction and variance schedule to initalize depthwise convolutional layers in PyTorch. `wconv` is the weight of a depthwise convolutional layer (nn.Conv2d), and $d \in [0, 1]$ is its depth as a fraction of the total depth.

# E SHIFT FUNCTION DEFINITION & PROOF

For a given matrix $Z \in \mathbb{R}^{k \times k}$ (say, a Gaussian kernel centered at 0, 0—the top left of the filter), we assume the shift operator is defined as follows:

$$\text{Shift}(Z, \delta x, \delta y)_{i,j} = Z_{(i+\delta x) \bmod k, (j+\delta y) \bmod k}. \tag{11}$$

Then, if

$$[C_{i,j}] = \text{Shift}(Z_\sigma, i, j) \tag{12}$$

and the operation $(.)^B$ is defined by

$$\Sigma^B = \Sigma' \iff [\Sigma_{i,j}]_{\ell,m} = [\Sigma'_{\ell,m}]_{i,j}, \tag{13}$$

and

$$[C_{i,j}]_{\ell,m} = \text{Shift}(Z, i, j)_{\ell,m} = Z_{(i+\ell) \bmod k, (j+m) \bmod k} \tag{14}$$

$$[C_{\ell,m}]_{i,j} = \text{Shift}(Z, \ell, m)_{i,j} = Z_{(\ell+i) \bmod k, (m+j) \bmod k}, \tag{15}$$

this shows that $[C_{i,j}]_{\ell,m} = [C_{\ell,m}]_{i,j}$ for all $1 \leq i, j, \ell, m \leq k$ (i.e., $C$ is "block-symmetric"), which shows $C = C^B$.

# F  ADDITIONAL IMAGENET EXPERIMENTS

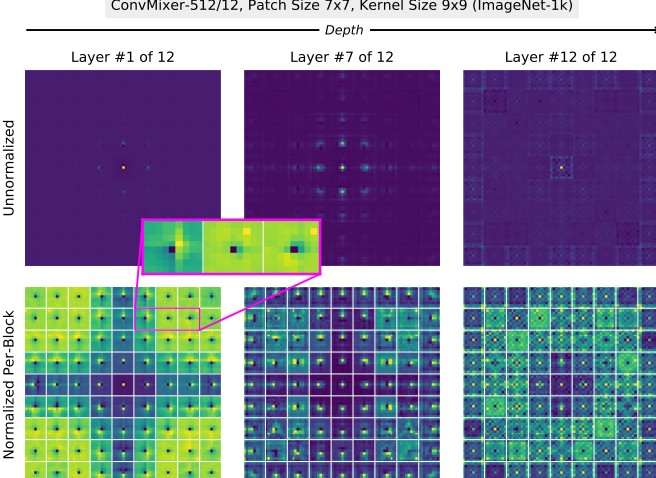

Figure 32: Covariance matrices from a ConvMixer trained on ImageNet exhibit similar structure to those of ConvMixers trained on CIFAR-10; however, later layers tend to have more structure, including a "checkerboard" pattern in each block.

## F.1 10-EPOCH IMAGENET RESULTS

| ConvMixer-512/12: Patch Size 14, Kernel Size 9 | Thawed | Frozen |
|---|---|---|
| Uniform init | 54.5 | 47.4 |
| Stats from CM-512/12 | 55.5 | 53.4 |
| Stats from CM-64/12 | 55.2 | 52.7 |
| Filters transferred from CM-512/12 | 55.1 | 54.4 |
| Our init (.15, .3, .5) | 55.4 | 52.2 |
| Our init (.15, .5, .25) | 55.5 | 52.4 |

Table 2: ConvMixer performance on ImageNet-1k training with 10 epochs. Our initialization performs comparably to loading covariance matrices from previously-trained models (which were trained for 150 epochs).

| ConvMixer-512/12: Patch Size 7, Kernel Size 9 | Thawed | Frozen |
|---|---|---|
| Uniform init | 61.87 | 56.73 |
| Stats from CM-512/12 | 62.56 | 60.79 |
| Stats from CM-64/12 | 62.72 | 60.86 |
| Filters transferred from CM-512/12 | 62.81 | 61.83 |
| Our init (.15, .3, .5) | 62.49 | 58.94 |
| Our init (.15, .5, .25) | 62.59 | 59.31 |

Table 3: ImageNet 10-epoch training

| ConvMixer-512/24: Patch Size 14, Kernel Size 9 | Thawed | Frozen |
|---|---|---|
| Uniform init | 50.40 | 43.00 |
| Stats from CM-512/12 | 53.03 | 51.45 |
| Stats from CM-64/12 | 53.16 | 51.25 |
| Filters transferred from CM-512/12 | 52.87 | 52.12 |
| Our init (.15, .3, .5) | 53.80 | 51.16 |
| Our init (.15, .5, .25) | 53.76 | 50.81 |

Table 4: ImageNet 10-epoch training

| ConvNeXt-Atto | Thawed | Frozen |
|---|---|---|
| Uniform init | 31.37 | 23.63 |
| Stats from the same arch | 33.44 | 40.41 |
| Stats from $1/8^{th}$-width arch | 29.81 | 31.47 |
| Filters transferred from same arch | 31.68 | 40.48 |
| Our init (.15, .3, .5) | 37.64 | 34.59 |
| Our init (.15, .5, .25) | 31.34 | 34.23 |
| Our init (.15, .25, 1.0) | 38.01 | 33.98 |

Table 5: ImageNet 10-epoch training

| ConvNeXt-Tiny | Thawed | Frozen |
|---|---|---|
| Uniform init | 32.51 | 25.94 |
| Stats from the same arch | 42.78 | 41.54 |
| Stats from $1/8^{th}$-width arch | 44.60 | 42.86 |
| Filters transferred from same arch | 31.01 | 45.32 |
| Our init (.15, .3, .5) | 35.64 | 35.04 |
| Our init (.15, .5, .25) | 40.17 | 38.91 |
| Our init (.15, .25, 1.0) | 40.78 | 36.62 |

Table 6: ImageNet 10-epoch training

## F.2 50-EPOCH IMAGENET RESULTS

| **ConvNeXt-Atto** | Thawed | Frozen |
|---|---|---|
| Uniform init | 69.96 | 51.43 |
| Stats from the same arch | 68.83 | 66.71 |
| Stats from $1/8^{\text{th}}$-width arch | 68.69 | 66.31 |
| Filters transferred from same arch | 68.01 | 67.29 |
| Our init (.15, .3, .5) | 65.55 | 63.48 |
| Our init (.15, .5, .25) | 67.84 | 64.52 |
| Our init (.15, .25, 1.0) | 68.06 | 63.43 |

Table 7: ImageNet **50-epoch** training

| **ConvMixer-512/12: Patch Size 14, Kernel Size 9** | Thawed | Frozen |
|---|---|---|
| Uniform init | 67.03 | 60.47 |
| Stats from the same arch | 67.13 | 65.08 |
| Stats from $1/8^{\text{th}}$-width arch | 66.75 | 64.94 |
| Filters transferred from same arch | 67.28 | 66.11 |
| Our init (.15, .3, .5) | 66.12 | 64.39 |
| Our init (.15, .5, .25) | 67.41 | 64.43 |
| Our init (.15, .25, 1.0) | 67.34 | 64.12 |

Table 8: ImageNet **50-epoch** training

| **ConvMixer-512/24: Patch Size 14, Kernel Size 9** | Thawed | Frozen |
|---|---|---|
| Uniform init | 67.76 | 62.50 |
| Stats from the same arch | 68.92 | 67.91 |
| Stats from $1/8^{\text{th}}$-width arch | 68.78 | 67.36 |
| Filters transferred from same arch | 69.42 | 68.66 |
| Our init (.15, .3, .5) | 69.05 | 66.20 |
| Our init (.15, .5, .25) | 69.60 | 66.57 |
| Our init (.15, .25, 1.0) | 69.52 | 66.38 |

Table 9: ImageNet **50-epoch** training

# G  ADDITIONAL CIFAR-10 TABLES

Table 10: CIFAR-10 results for ConvMixer-256/8 with patch size 2. **Bold** denotes the highest per group, and **blue bold** denotes the second highest.

| Filt. Size | # Eps | THAWED | | | | | FROZEN | | | | |
|---|---|---|---|---|---|---|---|---|---|---|---|
| | | Uniform | Cov. transfer | Cov. transfer ($\frac{1}{8}$ width) | Direct transfer | Our init | Uniform | Cov. transfer | Cov. transfer ($\frac{1}{8}$ width) | Direct transfer | Our init |
| 3 | 20 | 89.69 ± .20 | **90.08 ± .39** | 89.76 ± .14 | **90.02 ± .14** | 89.64 ± .13 | 88.99 ± .41 | **89.56 ± .11** | 89.54 ± .40 | **89.94 ± .08** | 88.94 ± .25 |
| | 50 | **91.92 ± .03** | **92.01 ± .21** | 91.83 ± .02 | 91.69 ± .18 | 91.63 ± .07 | 91.12 ± .22 | **91.42 ± .15** | 91.34 ± .27 | **91.62 ± .17** | 91.02 ± .25 |
| | 200 | **93.08 ± .16** | 92.94 ± .26 | 92.92 ± .19 | **92.97 ± .18** | 92.84 ± .11 | 92.38 ± .27 | **92.55 ± .09** | 92.27 ± .21 | **92.47 ± .19** | 92.09 ± .13 |
| 7 | 20 | 89.66 ± .21 | 89.91 ± .09 | **90.63 ± .10** | 90.48 ± .26 | **90.79 ± .24** | 86.73 ± .27 | 88.86 ± .36 | 89.36 ± .09 | **90.00 ± .27** | **90.22 ± .03** |
| | 50 | 91.81 ± .15 | 91.77 ± .26 | 91.88 ± .19 | **91.91 ± .20** | **92.48 ± .08** | 89.44 ± .23 | 90.72 ± .13 | 91.05 ± .12 | **91.54 ± .37** | **92.02 ± .23** |
| | 200 | **92.86 ± .15** | 92.77 ± .28 | 92.74 ± .15 | 92.72 ± .06 | **93.40 ± .20** | 90.70 ± .12 | 91.68 ± .16 | 91.84 ± .05 | **92.36 ± .20** | **92.83 ± .24** |
| 9 | 20 | 89.26 ± .35 | 89.70 ± .06 | **90.22 ± .36** | 90.14 ± .06 | **90.85 ± .14** | 84.97 ± .03 | 88.18 ± .12 | **89.67 ± .07** | 89.50 ± .37 | **90.56 ± .09** |
| | 50 | 91.74 ± .12 | 91.63 ± .15 | **92.11 ± .18** | 91.79 ± .18 | **92.54 ± .16** | 88.25 ± .39 | 90.22 ± .09 | 91.01 ± .12 | **91.23 ± .13** | **91.96 ± .12** |
| | 200 | 92.65 ± .16 | 92.53 ± .13 | **92.85 ± .12** | 92.62 ± .26 | **93.21 ± .09** | 90.04 ± .33 | 91.34 ± .08 | 92.00 ± .18 | **92.05 ± .28** | **93.00 ± .05** |
| 15 | 20 | 86.64 ± .30 | 88.19 ± .51 | **89.27 ± .05** | 89.17 ± .19 | **90.33 ± .22** | 81.99 ± .21 | 86.31 ± .15 | 87.52 ± .23 | **88.39 ± .28** | **90.38 ± .06** |
| | 50 | 89.94 ± .45 | 90.26 ± .16 | **90.81 ± .26** | 90.79 ± .24 | **92.17 ± .23** | 85.05 ± .33 | 88.71 ± .08 | 89.51 ± .06 | **90.08 ± .09** | **92.11 ± .24** |
| | 200 | 91.79 ± .23 | 91.60 ± .21 | **92.01 ± .19** | 91.87 ± .26 | **92.94 ± .11** | 87.64 ± .05 | 89.93 ± .25 | 90.61 ± .17 | **90.94 ± .12** | **93.02 ± .09** |

Table 11: CIFAR-10 results for ConvMixer-256/24 with patch size 2. **Bold** denotes the highest per group, and **blue bold** denotes the second highest.

| Filt. Size | # Eps | THAWED | | | | | FROZEN | | | | |
|---|---|---|---|---|---|---|---|---|---|---|---|
| | | Uniform | Cov. transfer | Cov. transfer ($\frac{1}{8}$ width) | Direct transfer | Our init | Uniform | Cov. transfer | Cov. transfer ($\frac{1}{8}$ width) | Direct transfer | Our init |
| 3 | 20 | 88.37 ± .11 | 88.79 ± .07 | **88.97 ± .13** | **89.22 ± .26** | 88.42 ± .14 | 87.74 ± .32 | 88.63 ± .20 | **88.80 ± .12** | **88.95 ± .11** | 88.05 ± .09 |
| | 50 | 92.38 ± .15 | 92.30 ± .24 | **92.46 ± .07** | **92.56 ± .19** | 92.19 ± .24 | 91.86 ± .18 | 92.02 ± .32 | **92.37 ± .03** | **92.31 ± .14** | 92.00 ± .23 |
| | 200 | 94.13 ± .07 | 94.32 ± .12 | **94.41 ± .11** | **94.37 ± .03** | 94.16 ± .20 | 93.71 ± .15 | 93.91 ± .13 | **94.28 ± .20** | **93.95 ± .23** | 93.84 ± .10 |
| 7 | 20 | 88.49 ± .46 | 89.08 ± .15 | **89.90 ± .14** | 89.81 ± .16 | **90.28 ± .18** | 85.81 ± .05 | 87.98 ± .08 | 89.19 ± .12 | **89.34 ± .40** | **90.09 ± .22** |
| | 50 | 91.90 ± .17 | 91.73 ± .26 | **92.39 ± .17** | 92.25 ± .12 | **93.15 ± .08** | 89.94 ± .17 | 90.86 ± .07 | 91.56 ± .09 | **91.91 ± .08** | **92.80 ± .27** |
| | 200 | 93.57 ± .08 | 93.43 ± .21 | **93.71 ± .20** | 93.62 ± .19 | **94.44 ± .26** | 91.78 ± .22 | 92.78 ± .16 | **93.24 ± .25** | 93.00 ± .25 | **94.03 ± .23** |
| 9 | 20 | 87.99 ± .41 | 88.25 ± .13 | **89.44 ± .03** | 89.42 ± .26 | **90.54 ± .15** | 83.42 ± .33 | 87.38 ± .18 | 88.46 ± .45 | **88.90 ± .52** | **90.31 ± .12** |
| | 50 | 91.06 ± .11 | 91.36 ± .17 | **91.87 ± .11** | 91.69 ± .06 | **93.03 ± .10** | 88.38 ± .08 | 90.25 ± .34 | **91.12 ± .16** | 91.04 ± .18 | **92.78 ± .27** |
| | 200 | 93.12 ± .37 | 93.08 ± .17 | **93.42 ± .21** | 93.16 ± .21 | **94.12 ± .18** | 90.86 ± .07 | 91.86 ± .28 | **92.60 ± .09** | 92.53 ± .15 | **94.03 ± .09** |
| 15 | 20 | 84.95 ± .50 | 85.80 ± .48 | 86.67 ± .29 | **87.69 ± .57** | **90.08 ± .23** | 80.72 ± .20 | 84.27 ± .41 | 85.70 ± .25 | **86.75 ± .52** | **90.03 ± .06** |
| | 50 | 89.74 ± .11 | 89.68 ± .15 | 90.10 ± .12 | **90.22 ± .27** | **92.30 ± .16** | 85.10 ± .27 | 88.05 ± .11 | 88.78 ± .33 | **89.72 ± .21** | **92.93 ± .24** |
| | 200 | 92.03 ± .10 | 92.02 ± .23 | **92.22 ± .18** | 92.20 ± .03 | **93.66 ± .33** | 88.18 ± .16 | 90.19 ± .32 | 90.67 ± .18 | **91.19 ± .20** | **94.16 ± .12** |

Table 12: CIFAR-10 results for ConvMixer-256/8 with patch size 1. **Bold** denotes the highest per group, and **blue bold** denotes the second highest.

| Filt. Size | # Eps | THAWED | | | | | FROZEN | | | | |
|---|---|---|---|---|---|---|---|---|---|---|---|
| | | Uniform | Cov. transfer | Cov. transfer ($\frac{1}{8}$ width) | Direct transfer | Our init | Uniform | Cov. transfer | Cov. transfer ($\frac{1}{8}$ width) | Direct transfer | Our init |
| 3 | 20 | 90.41 ± .11 | 90.60 ± .31 | **90.78 ± .22** | **90.66 ± .18** | 89.84 ± .27 | 89.39 ± .09 | **90.37 ± .06** | 90.14 ± .03 | **90.64 ± .17** | 89.07 ± .21 |
| | 50 | 91.89 ± .06 | 92.00 ± .23 | **92.09 ± .18** | **92.05 ± .20** | 91.49 ± .10 | 90.97 ± .19 | **91.87 ± .09** | 91.65 ± .10 | **91.90 ± .26** | 90.73 ± .26 |
| | 200 | 92.16 ± .19 | 92.38 ± .12 | **92.58 ± .09** | **92.54 ± .18** | 91.85 ± .15 | 91.71 ± .20 | **92.12 ± .08** | **92.05 ± .36** | 91.98 ± .16 | 91.47 ± .14 |
| 7 | 20 | 91.70 ± .16 | 91.94 ± .08 | **92.02 ± .05** | **92.37 ± .07** | 91.80 ± .23 | 89.84 ± .04 | 90.78 ± .27 | **91.30 ± .11** | **91.92 ± .03** | 91.08 ± .15 |
| | 50 | 93.30 ± .15 | 93.00 ± .25 | 93.31 ± .10 | **93.42 ± .07** | **93.33 ± .29** | 91.51 ± .20 | 92.21 ± .06 | **92.68 ± .07** | **93.01 ± .02** | 92.31 ± .32 |
| | 200 | 93.67 ± .13 | 93.59 ± .08 | 93.68 ± .12 | **93.81 ± .23** | **93.83 ± .18** | 92.48 ± .16 | 92.89 ± .18 | 92.98 ± .05 | **93.38 ± .04** | **93.16 ± .20** |
| 9 | 20 | 91.92 ± .16 | 91.74 ± .25 | 92.07 ± .19 | **92.21 ± .17** | **92.37 ± .19** | 89.45 ± .02 | 90.41 ± .24 | 91.25 ± .16 | **91.85 ± .03** | **91.35 ± .05** |
| | 50 | 93.25 ± .23 | 92.92 ± .05 | 93.22 ± .16 | **93.37 ± .01** | **93.45 ± .19** | 91.13 ± .13 | 91.93 ± .17 | **92.52 ± .03** | **92.85 ± .16** | 92.45 ± .27 |
| | 200 | 93.74 ± .35 | 93.60 ± .16 | 93.68 ± .14 | **93.83 ± .03** | **94.12 ± .24** | 91.96 ± .07 | 92.67 ± .04 | 93.08 ± .17 | **93.38 ± .01** | **93.56 ± .14** |
| 15 | 20 | 90.65 ± .23 | 91.00 ± .05 | **91.91 ± .17** | 91.70 ± .28 | **92.29 ± .21** | 86.64 ± .14 | 89.09 ± .32 | 90.70 ± .07 | **91.32 ± .10** | **91.53 ± .12** |
| | 50 | 92.76 ± .06 | 92.54 ± .10 | **92.99 ± .13** | 92.95 ± .10 | **93.42 ± .07** | 89.20 ± .17 | 90.65 ± .15 | 91.72 ± .23 | **92.35 ± .13** | **92.73 ± .07** |
| | 200 | 93.55 ± .16 | 93.22 ± .16 | 93.57 ± .03 | **93.59 ± .02** | **94.13 ± .13** | 90.20 ± .22 | 91.83 ± .14 | 92.79 ± .17 | **93.01 ± .27** | **93.47 ± .05** |

Table 13: CIFAR-10 results for ConvNeXt-atto with patch size 1. **Bold** denotes the highest per group, and **blue bold** denotes the second highest.

| Filt. Size | # Eps | THAWED | | | | | FROZEN | | | | |
| --- | --- | --- | --- | --- | --- | --- | --- | --- | --- | --- | --- |
| | | Uniform | Cov. transfer | Cov. transfer ($\frac{1}{8}$ width) | Direct transfer | Our init | Uniform | Cov. transfer | Cov. transfer ($\frac{1}{8}$ width) | Direct transfer | Our init |
| 7 | 20 | $81.46 \pm .51$ | $85.00 \pm .21$ | $84.71 \pm .25$ | $86.18 \pm .59$ | $86.44 \pm .78$ | $80.45 \pm .88$ | $83.91 \pm .13$ | $84.06 \pm .19$ | $86.12 \pm .57$ | $84.71 \pm .08$ |
| | 50 | $87.55 \pm .46$ | $88.38 \pm .10$ | $87.92 \pm .27$ | $89.07 \pm .75$ | $90.54 \pm .16$ | $85.18 \pm .39$ | $87.81 \pm .10$ | $86.92 \pm .37$ | $89.32 \pm .21$ | $90.13 \pm .14$ |
| | 200 | $90.47 \pm .68$ | $90.77 \pm .46$ | $90.69 \pm .11$ | $91.05 \pm .40$ | $92.03 \pm .35$ | $87.10 \pm .08$ | $89.70 \pm .21$ | $89.08 \pm .24$ | $90.83 \pm .21$ | $92.27 \pm .22$ |

Table 14: CIFAR-10 results for ConvNeXt-atto with patch size 2. **Bold** denotes the highest per group, and **blue bold** denotes the second highest.

| Filt. Size | # Eps | THAWED | | | | | FROZEN | | | | |
| --- | --- | --- | --- | --- | --- | --- | --- | --- | --- | --- | --- |
| | | Uniform | Cov. transfer | Cov. transfer ($\frac{1}{8}$ width) | Direct transfer | Our init | Uniform | Cov. transfer | Cov. transfer ($\frac{1}{8}$ width) | Direct transfer | Our init |
| 7 | 20 | $72.96 \pm .70$ | $82.36 \pm .32$ | $81.29 \pm .78$ | $83.42 \pm .47$ | $81.13 \pm .54$ | $72.18 \pm .30$ | $80.82 \pm .17$ | $79.03 \pm .07$ | $83.61 \pm .23$ | $79.36 \pm .07$ |
| | 50 | $83.71 \pm .60$ | $86.21 \pm .21$ | $86.32 \pm .16$ | $86.28 \pm .20$ | $85.97 \pm .24$ | $77.92 \pm .34$ | $85.16 \pm .45$ | $84.83 \pm .09$ | $85.93 \pm .20$ | $85.45 \pm .20$ |
| | 200 | $86.28 \pm .02$ | $87.40 \pm .27$ | $88.14 \pm .18$ | $87.02 \pm .10$ | $88.20 \pm .52$ | $79.82 \pm .38$ | $86.25 \pm .10$ | $86.60 \pm .02$ | $86.59 \pm .33$ | $87.69 \pm .28$ |

