# OpenReview forum: "Understanding the Covariance Structure of Convolutional Filters"
_ICLR.cc/2023/Conference — ICLR 2023 poster_

### Official Review · Reviewer_Yta9 · 2022-10-24

**Confidence:** 4
**Correctness:** 3
**Technical Novelty And Significance:** 3
**Empirical Novelty And Significance:** 2
**Recommendation:** 6

**Clarity, Quality, Novelty And Reproducibility:**

The paper is well written and easy to read. I have a few questions:
- What is a sub-block as mentioned in Figure 1?
- That the notation of [ ] in equation 3 means? Are you trying to estimate the covariance Simga_i
 from H vectors?
- How do you directly transfer CM-256/24 to CM-256/8 in Figure 3 (D in first subplot)? It is not very clear to me.
- Do you observe edge-like filters in the first layer of CNNs? If so, are those filters look very different to the ones built from the multivariate initialization scheme? Could that explain the performance gap between Thawed and Frozen?
- There a few typos in the text, such as in the caption of Fig 4, the paragraph of eq (6).
- What does ‘free’ mean in the last sentence of the conclusion?

**Strength And Weaknesses:**

This paper builds upon an interesting observation of the covariance structures of each convolution filter at multiple layers of CNNs. It proposes a parametric model for such covariance structures such that certain parameters may be fitted from trained models.

Certain results of this paper could still be improved to support the point of the learning-free multivariate initialization scheme. It is still not very clear on the ImageNet case, if the learning-free approach could achieve a similar performance compared to the learning approach. In Table 1 Frozen vs. Thawed, we see that the results with the Frozen case are worse than those of the Thawed case. It is thus not clear why learning is not needed. One suggestion would be to fit the parametric model of covariances to some small ImageNet models, and then to see if they work well (as CIFAR is quite different to ImageNet, using the parameters from CIFAR might have some unpredictable risk).

**Summary Of The Paper:**

This paper studies the covariance structures of each convolution filter in deep learning models for classification problems. It builds a parametric model for these structures and proposes to directly sample a multivariate Gaussian distribution as an initialization for a family of CNNs. Numerical results show that in some cases, such initialization may achieve very high classification, without the need of learning.

**Summary Of The Review:**

REVISED: I have raised my score to 6 as I still find the significance of the paper could be improved by providing further discussion on the impact of the initialization of other layers beyond depthwise convolutional layers . This is not so clear by reading the paper. Nevertheless, the authors proposed an interesting method to model the covariance structure of these layers, which gives an explanation of the performance of CNN. This is novel and could be published.

---

> ### Author Response · Authors · 2022-11-15
> **Clarifications, new experiments**
>
> Thanks for your review!
>
> Unfortunately, I'm not really following your point here. I think you may be conflating our terms "frozen" and "learning-free". When we talk about "frozen" filters in the paper, we mean that the convolutional filters do not receive gradient updates during training. When we mention that our technique is "learning-free", we mean that the initialization just involves random sampling from a Gaussian distribution that we specified, more or less, by hand (i.e., it doesn't rely on analyzing the dataset in any way).
>
> Generally, you would expect that a CNN would achieve poor performance if the filters were not trained. But we show that our initialization is so effective that we can achieve reasonable performance (and sometimes even superior performance!) if we don't train the filters at all after initializing with our technique. So we are not generally asking "why isn't the frozen case working as well as the thawed case?", because we think it's very surprising that the frozen case _ever_ works as well as the thawed case.
>
> Please let me know if I've misunderstood your point -- I'm happy to discuss more.
>
> In any case, we have added more experiments in the rebuttal revision: we have added other initialization techniques as baselines, and our technique outperforms them. We have also demonstrated our technique on three new datasets: CIFAR-100, SVHN, and EuroSAT. This provides more evidence for the strength of our initialization method -- while the benefits are sometimes small for the easy SVHN and EuroSAT datasets, they are much more significant for the harder CIFAR-100 dataset. Please see Appendix B of our rebuttal revision for these results.
>
> ### Clarity, etc.
>
> * We are referring to the blocks of our representation of the covariance matrix; each block i, j represents the covariance of pixel i, j in the filter with all other pixels in the filter. We now see there is actually no reason to call this a _sub-_block, and just calling it a _block_ would be sufficient; we'll clear this up in the paper.
>
> * Yes, we are estimating the covariance from the $F$ vectors.  Would it be more clear if we used parentheses instead of square brackets?
>
> * Ah, this is a typo... the transfers in the first and last plots of Figure 3 are from models of the same depth (it should be, e.g., "Cov. transfer from CM-256/8"). We'll fix this. In _Figure 4_, we investigate transfers from differently-sized models. Thanks for catching this.
>
> * The first layer of ConvMixer and ConvNeXt is a patch embedding layer, which is typically implemented as a convolution with equal stride and kernel size. We did not include this in our initialization technique (we only considered the "internal", large-kernel depthwise convolutions). On models using large patches (e.g., ImageNet models), these patch embedding filters look quite different from the internal filters, and are definitely harder to model. They are Gabor-like filters (somewhat more like the well-defined edge detectors you may be alluding to), and are not easily-described by a single Gaussian. We found that a mixture of Gaussians (with, e.g., n = 5) can more effectively describe them -- but this is not amenable to a nice, closed-form representation like our method in the paper.
>
>     If you're just referring to the first layer of depthwise convolutions, these don't actually look very different from those in our initialization scheme. We have included a new page of graphics in our revision to show the actual filters vs. filters from our initialization for a variety of layers (please see Appendix A).
>
> * Thanks, we fixed the typo in Fig. 4, but do not see the typo you mentioned around Eq. 6.
>
> * We mean that our initialization requires practically no compute -- building the covariance matrix is very simple in NumPy, and sampling from the corresponding Gaussian to get filters is also very cheap. That is, our initialization is practically no more expensive than uniform initialization. We have also shown that the hyperparameters of the method are widely applicable, so there is likely no need to search for them for a new application (in our new experiments in Appendix B, we used the same hyperparameters as for our CIFAR-10 experiments).
>
> Looking forward to hopefully discussing more, and thanks again for your time!

---

> > ### Comment · Reviewer_Yta9 · 2022-11-15
> > **1+ question**
> >
> > Thanks for your clarification. If I understood correctly, you did not apply the initlization scheme to the patch embedding layer as it is not easy to be described by the proposed covariance model. So how do you initialize the filters in this layer, by hand / without learning ?

---

> > > ### Author Response · Authors · 2022-11-15
> > > **Initialization of other layers**
> > >
> > > All the layers except _depthwise_ convolutions were initialized in the standard way, i.e., Kaiming uniform initialization. So the patch embedding layer was simply uniformly initialized from $\mathcal{U}(-\tfrac{1}{\sqrt{3}p}, \tfrac{1}{\sqrt{3}p})$,  where 3 is the number of input channels to the model and $p$ is the patch size. Note the patch embedding layer is not a depthwise convolution. And in the majority of our experiments, it also has a very small "filter size". Our custom initialization technique is only meant to be applied to depthwise convolutional layers -- we think this is clear in the paper, but please let us know if you have other thoughts.

---

> ### Author Response · Authors · 2022-11-26
> **Improving clarity**
>
> Thank you again for engaging with our paper and updating your score. We have some additional thoughts regarding your revised summary.
>
> First, with your feedback in mind, we updated the "Scope" section on page 2 to make it more clear that our study is restricted to large-filter depthwise convolutional layers. The paragraph now gives a brief overview of the ConvMixer and ConvNeXt architectures which distinguishes our focus on depthwise convoutional layers, as opposed to, e.g., patch embeddings (which can be viewed as a type of strided, normal convolution).
>
> But respectfully, we think our paper is quite clear on the fact that we are interested only in initializing (large-filter) depthwise convolutional layers, and not other layers. For example, the first paragraph of the introduction juxtaposes "large-filter convolutions" with "stacked small-filter convolutions", and goes on to mention that the separate processing of spatial- and channel- dimensions is what allows our study: _"These new models also completely separate the processing of the channel and spatial dimensions, meaning that the now-single-channel filters are, in some sense, more independent from each other than in previous models such as ResNets"_. That is, we believe the first paragraph focuses our study on depthwise convolutional filters; we are saying that their large, structured filters and independent processing of channels is what enables the study. Admittedly, we could have said "depthwise" explicitly in this paragraph -- though we did say it twice in the abstract. We will make this change, for example:
>
> "These new models also completely separate the processing of the channel and spatial dimensions, meaning that the now-single-channel filters are, in some sense, more independent from each other than in previous models such as ResNets" $\longrightarrow$ "**These new models use _depthwise convolution_, completely separating the processing of the channel** and spatial dimensions, meaning that the now-single-channel filters are, in some sense, more independent from each other than in previous models such as ResNets"
>
> We also say clearly in "Preliminaries" that we're only interested in depthwise convolutional filters, and we discuss them in the paragraph "Basic method." of Section 2, in addition to numerous other places.
>
> While we are no longer able to update the paper, we're glad to further clear up any language that might make the scope of our study unclear, posting the edits here. Beyond the addition of the keyword "depthwise" in the introduction, we are unsure how to make this more clear (we think it is already very clear, to be honest).
>
> -------------------
>
> On a separate note, we have also added more experiments in our final revision. We added experiments on Tiny ImageNet, where our initialization also leads to strong performance gains (importantly, with no additional tuning). And we demonstrated the impact of our initialization on data efficiency (using CIFAR-10) -- here, in the low-data setting, our initialization leads to particularly large increases in accuracy.

---

> ### Author Response · Authors · 2022-12-05
> **Discussion ends on Dec. 12; we would appreciate hearing back from you again**
>
> In our previous post below, we tried to address your concerns about "providing further discussion on the impact of the initialization of other layers beyond depthwise convolutional layer". We're honestly not 100% sure what you meant, but we assumed that our paper was not clear enough about its focus on _specifically_ depthwise convolutional layers. In our revision, we modified "Scope" to make this clearer, and we also proposed some clarifying changes in our earlier comment.
>
> If you were asking about how our initialization _interacts with_ the initialization of other layers, e.g., the pointwise (linear) layers: we did not study this. All layers except depthwise convolutional layers were initialized in the standard way, i.e., Kaiming uniform initialization.
>
> If you were asking about how our initialization would perform when used on, e.g., normal convolutional layers: we don't have much evidence for this at the moment. But our initialization is specifically meant for large-kernel convolutions, and normal convolutional layers typically use 3x3 convolutions. We are very forthright in the paper that our technique does not offer an improvement for 3x3 convolutions, as they have practically no structure to model. We aren't aware of any (modern) architectures that use large-kernel plain convolutions (when they _do_ use large-kernel convolutions, it's usually just for ~1 of the first layers).
>
> If you were asking if our technique can be applied to other types of layers, such as pointwise (linear) layers: the closed-form initialization technique we proposed is crafted for depthwise convolutional layers only, but the _covariance transfer_ concept _could_ be applied to other types of layers. We have investigated this for pointwise (i.e., linear) layers, but have not been able to get any interesting results. These layers have no discernible covariance structure.
>
> Essentially, if your concern is that our paper wasn't clear enough about its focus on depthwise convolution, we believe that's now fixed.
>
> Thanks again for your time.

---

> > ### Comment · Reviewer_Yta9 · 2022-12-07
> > **clarification**
> >
> > I think that you have made things more clear now in the paper. I was wondering about the how your proposed initialization interacts with the initialization of other layers, to understand better the classfication accuracy as an ablation study.  Anyway, I have revised the summary of the review. I think the impact could be larger if the technique can be applied to other types of layers.

---

> > > ### Author Response · Authors · 2022-12-08
> > > **Some experiments**
> > >
> > > Thanks for your response. We have done some experiments to investigate how our initialization interacts with the initialization of other layers.
> > > We initialized all pointwise layers orthogonally or with Xavier/Kaiming normal weights (as opposed to the previous setting, which only used Kaiming uniform initialization).
> > > Then, we either used the corresponding initialization for depthwise convolution (orthogonal or Xavier/Kaiming normal -- labeled **Without** in the tables below)
> > > or *our* initialization (labeled **With**). We present the results below for the three initializations.
> > > Note that our initialization results in a fairly large boost in accuracy regardless of the way the pointwise layers are initialized.
> > > The experiments are on CIFAR-10 using a ConvMixer-256/8 with 2x2 patches and 7x7 filters.
> > > (We haven't been able to replicate each experiment thrice as usual yet, but we think the effect sizes are convincing -- we also only present thawed filters.)
> > >
> > > | | Orthogonal Init | | |
> > > |--|--|--|--|
> > > |Epochs|Without Our Init|**With Our Init**| Improvement |
> > > |20|89.28|90.55| +1.26 |
> > > |50|91.57|92.38| +0.81 |
> > > |200|92.78|93.25| +0.47 |
> > >
> > > | | Xavier Normal Init | | |
> > > |--|--|--|--|
> > > |Epochs|Without Our Init|**With Our Init**| Improvement |
> > > |20|89.52|90.60| +1.08 |
> > > |50|91.59|92.66| +1.07 |
> > > |200|92.40|92.89| +0.49 |
> > >
> > > | | Kaiming Normal Init | | |
> > > |--|--|--|--|
> > > |Epochs|Without Our Init|**With Our Init**| Improvement |
> > > |20|88.33|90.69| +2.36 |
> > > |50|90.95|92.08| +1.13 |
> > > |200|92.40|92.92| +0.52 |
> > >
> > > Note that the improvement from using our initialization is about the same (given the same number of epochs) for orthogonal, Kaiming normal, and Xavier normal.
> > >
> > > As an aside, we didn't use Dirac init for the pointwise layers because we found that the models do not train very well when the pointwise layers are initialized that way (regardless of using our initialization or others).
> > >
> > > We think it's intuitive that our initialization would be beneficial regardless of the initialization of the other layers,
> > > so long as that initialization allows for training the network effectively.
> > >
> > > No doubt the method would have more impact if it could be applied to other types of layers (i.e., beyond convolutional ones). But it doesn't seem like pointwise (or linear) layers have any structure to analyze as we did in the paper -- we have spent lots of time investigating this. We think that such a technique would be highly nontrival, must likely account for permutation symmetries, and would be worthy of a new paper. As it stands, our technique for just depthwise convolutional layers already appears to be very useful for the increasingly-popular ViT-style modern CNNs. It reduces training time, improves the final accuracy, and even improves data efficiency -- all for practically zero additional overhead. We think this is a solid first step for "smart" initialization techniques that may be able to reduce the burden of large-scale pretraining.

---

> > > > ### Comment · Reviewer_Yta9 · 2022-12-12
> > > > **summary**
> > > >
> > > > Dear authors,
> > > >
> > > > Thanks for the additinoal results. Could you summarize, in a few lines, your main findings / messages of these results ? For example, in what situations / examples,
> > > > -  you initialization is beneficial in the thawed case compared to existing ones ?
> > > > -  the frozen case works as well as the thawed case ?
> > > > -  the frozen case does not work as well as the thawed case ?
> > > >
> > > > best,

---

> > > > > ### Author Response · Authors · 2022-12-12
> > > > > **Response**
> > > > >
> > > > > The results we posted above are only for the thawed rather than the frozen case, so we ran the frozen case  too so we could better answer your question.
> > > > >
> > > > > | | Orthogonal Init (Frozen) | | |
> > > > > |--|--|--|--|
> > > > > |Epochs|Without Our Init|**With Our Init**|Improvement|
> > > > > |20|86.82|89.74|+2.92|
> > > > > |50|90.41|91.60|+1.19|
> > > > > |200|92.13|92.91|+0.78|
> > > > >
> > > > >
> > > > > | | Xavier Normal Init (Frozen) | | |
> > > > > |--|--|--|--|
> > > > > |Epochs|Without Our Init|**With Our Init**|Improvement|
> > > > > |20|87.09|89.62|+2.53|
> > > > > |50|89.29|91.52|+2.23|
> > > > > |200|91.04|92.49|+1.45|
> > > > >
> > > > > | | Kaiming Normal Init (Frozen) | | |
> > > > > |--|--|--|--|
> > > > > |Epochs|Without Our Init|**With Our Init**|Improvement|
> > > > > |20|85.28|89.79|+4.51|
> > > > > |50|88.48|91.64|+3.16|
> > > > > |200|90.30|92.67|+2.37|
> > > > >
> > > > > So with respect to the initialization interaction results above, here are our answers:
> > > > >
> > > > > - Our initialization is beneficial in _all cases investigated_.
> > > > > - The frozen case doesn't work as well as the thawed case in these experiments, though it comes within ~0.3% in the 200-epoch trials. However, the frozen case _with our init_ **does better** than the thawed case _without our init_ for Kaiming Normal, Orthogonal, 200- and 20-epoch Xavier Normal. That is, **our init does better than Orthogonal, Xavier, and Kaiming-initialized networks even when the resulting filters are frozen**.
> > > > > - Using our init, thawed does better than frozen in all the interaction experiments above. But the only case where our init (frozen) doesn't work as well as the default init is 50-epoch Xavier Normal (by a small amount).
> > > > >
> > > > > In summary, **our initialization shows strong performance gains when used within these differently-initialized networks, and this even applies if the filters are frozen after initialization**.
> > > > >
> > > > > ------------------------
> > > > >
> > > > > Also, we suspect that you might want the answers for all of our results, rather than just the newest ones in our replies to you. So here are those answers as well:
> > > > >
> > > > > - **Our initialization outperforms the existing ones in basically _all cases_ studied.** It outperforms uniform, dirac, orthogonal, and zero initialization on a variety of datasets. It even outperforms the "oracle" initialization of using pre-trained filters to initialize the model, on both ConvMixer and ConvNeXt. It's beneficial for ConvMixers and ConvNeXt on ImageNet, including two results for full-scale, full-duration training. Our initialization is especially beneficial in terms of achieving higher accuracy faster; it always results in substantially higher accuracy for short-duration training settings (e.g., 20 or 50 epochs instead of 200) -- see Figures 8 and 21. It's easier to list the cases where it didn't help: Our initialization _isn't_ beneficial for 3x3 or smaller filters (but it _is_ beneficial for 5x5 filters). It also wasn't beneficial for ConvNeXt-atto on ImageNet -- but it was _quite_ beneficial for ConvNeXt-tiny (atto is substantially smaller than tiny). This is consistent with our observation that our init most helps deep models.
> > > > >
> > > > > - The frozen case works as well as the thawed case generally for deeper models with larger kernel sizes. The most prominent examples are ConvMixer-256/24 and ConvNeXt-atto on CIFAR-10, as well as ConvMixer-512/16 on Tiny ImageNet. The frozen case also outperforms the thawed _uniform_ case on a practical-scale ImageNet model, ConvMixer-1024/32 (and comes close for ConvNeXt-Tiny). Note that the frozen case of our initialization is _always_ much better than the frozen case of _other_ initializations.  (And even when the frozen case doesn't work as well as the thawed, it is still typically competitive with other _thawed_ initializations.) The frozen case actually does better than the thawed case -- both using our init -- for ConvNeXt-atto (patch size 1x1) and ConvMixer-256/24 (kernel size 15x15) on CIFAR-10.
> > > > >
> > > > > - The frozen case doesn't work as well as the thawed case for shallower models (e.g., ConvMixer-256/8 on CIFAR-10/100 or ConvMixer-512/12 on ImageNet or ConvMixer-64/4 on SVHN). But the frozen case _always_ does surprisingly well, considering it's frozen (meaning the models have ~15-40% fewer learnable parameters, depending on the filter size). On our ImageNet results, while using our initialization and freezing the filters is (of course) not as effective as keeping them thawed, our initialization with _frozen_ filters sometimes (for deeper models) does better than _thawed_ Kaiming uniform initialization -- which we think is very surprising. (And this happens on Tiny ImageNet and CIFAR-10 as well.) The trend involving depth is quite apparent in Table 1.
> > > > >
> > > > > Thanks for your response -- we're happy to answer more questions, though the discussion period closes in less than a day.

---

### Official Review · Reviewer_tjLw · 2022-10-28

**Confidence:** 2
**Correctness:** 3
**Technical Novelty And Significance:** 3
**Empirical Novelty And Significance:** 4
**Recommendation:** 6

**Clarity, Quality, Novelty And Reproducibility:**

Clarity: The paper is well-written, but it can be improved by fixing the typos.

Quality: The effectiveness of the proposed initialization method with multiple experiments is experimentally sound. However, the paper lacks theoretical justifications of why the proposed covariance matrices perform well, other than the intuition and the observed pattern in trained networks.

Novelty: The idea of using pre-trained network weights for initialization is not novel. However, the proposed closed-form multivariate scheme is novel and it can be useful.

Reproducibility: The code is not provided, but the implementation details provided can be used to reproduce the results.



**Strength And Weaknesses:**

Strengths:
- Building a closed-form initialization scheme based on the observations from pre-trained networks is notable.
- The authors evaluate their method in variaous variants and the visual presentation of the results helps the reader to better see the differences across the variants.

Weaknesses:
- The closed-form covariance matrix computation is not completely learning-free, as the authors perform a grid search to find the optimal parameters.
- It is not clear why the authors use a quadratic schedule for the variance.
- The related works section can be improved by adding other recent initialization methods and comparing the results to them.


**Summary Of The Paper:**

This paper presents an initialization scheme for convolutional filters. The main idea is to first build a filter covariance matrix from pre-trained networks and then sample initial filters based on this covariance. The paper also provides a learning-free closed-form alternative for filter initialization. The experiments specifically focus on ConvMixer and ConvNext networks, and show that using the proposed initialization scheme outperforms uniform initialization. They also show that in some cases, networks with fixed frozen filters can achieve competitive performance.

**Summary Of The Review:**

The main idea of the paper is interesting and useful to the community.

Questions from the authors:
- It would be helpful to see the results of initializing with covariances from smaller models when two or more variables change simultaneously, for instance, from a model both narrower and shallower, to see if similar results will be obtained with more efficient pre-training.
- Would it be useful to use the proposed method for initializing networks with normal convolutions like ResNets?

---

> ### Author Response · Authors · 2022-11-15
> **New experiments, clarifications**
>
> Thanks for your review! We have run some more experiments comparing to other initialization techniques as you requested (Appendix B in the revision). We address your concerns point-by-point below.
>
> ### Weaknesses
>
> * Grid search: We're honestly unsure on this point. While we did do a grid search to find optimal parameters, the search is fairly decoupled from the actual experiments... for example, on CIFAR-10 the grid search was over ConvMixer-256/8, kernel size 9x9, patch size 2, trained for 20 epochs with _frozen filters_. We then used these parameters for much deeper models, _without_ frozen filters, trained for much longer, across a variety of kernel sizes. We even used the same parameters on CIFAR-100 (a new dataset), and saw significant performance improvements -- and we also saw decent performance on the SVHN and EuroSAT datasets (Appendix B), without re-tuning. So we don't think grid search is _necessary_, i.e., the one we did in the paper is in some sense "enough". Despite doing a grid search specifically for ConvNeXt, we actually used the parameters from the ConvMixer search, and these worked very well. The grid search was mostly to show that lots of parameters work (~99% of them result in _some_ improvement, at least in the 20-epoch setting). Similarly for ImageNet: we did a grid search for ConvMixer, in an even more limiting 10-epoch setting, and found that these worked for a variety of other architecture configurations as well as ConvNeXt. (We also talk about the variability in grid search results in our response to Reviewer JJAa.)
>
>     So we _did_ do a grid search, and that _could_ be called "learning", but if the parameters we found from this (very heuristic/limited) search are in some sense "globally okay" and we can use them on other datasets, is it not "learning-free"? Please let us know what you think -- we can fix this language in the paper if you still think it doesn't fit. But we think that "learning-free" is a fair claim, especially given our new results showing that hyperparameters transfer well to a variety of other datasets.
>
> * Just based on our visual inspection, it seemed like the "effective filter size" in the covariance matrices didn't grow linearly, but rather got rapidly larger towards the last couple layers. This seemed more quadratic than linear to us, so we included a quadratic term. But this choice is ultimately subjective, similarly to our choice to build our covariance matrices from Gaussian filters (others would have worked just as well, presumably).
>
> * We provide in Appendix B in our revision some evidence that our method is substantially better than uniform, dirac, zero, and orthogonal initialization on CIFAR-10 and CIFAR-100. Results are closer for the (much easier) SVHN and EuroSAT datasets, but we still do better or the same, particularly for short-duration training. We will try to provide results on another dataset such as Tiny ImageNet before the end of the rebuttal period.
>
> ### Clarity, etc.
>
> * We have fixed some typos that were pointed out (and we will do another pass before the rebuttal revision deadline).
>
> * Yes, this is very much an empirical paper. We would eventually like to put this concept on more sound theoretical footing. But for now, we have tried to "triangulate" the initialization scheme: our visual observations of covariances and our findings on "covariance transfer" pointed to the existence of such a technique, and we verified our technique works well across hundreds of experiments (with each trial repeated 3x when possible so we have a reasonable degree of statistical significance), on various datasets, architecture configurations, and training durations.
>
> * We have provided a more complete implementation in the appendix in the rebuttal revision.
>
> ### Questions
>
> * We agree it would be good to test covariance transfer from a model that differs in several dimensions, though we strongly suspect it would work as well as one or the other (at least for depth/width). We will attempt to try this, though we are currently putting more weight on testing our method on other datasets/against other baselines.
>
> * Our initialization technique makes the most sense for large-filter depthwise convolutions; in contrast, ResNet uses small-filter normal convolutions. The fact that these layers combine spatial and channel mixing makes it unclear if our technique is appropriate (and large-filter normal convolution is uncommon). However, we have tried covariance transfer in small ResNets, i.e., this one [0] used in the DAWNbench competition. While accuracy initially increased much faster, the final accuracy was approximately the same. We would have to do more testing to say for sure, but we suspect the method would not be quite as helpful. Also, we have looked at covariance matrices in the large-kernel (normal) convolutional layers of AlexNet, and found similar covariance structure to that found in the CNNs we investigated.
>
> [0] https://github.com/davidcpage/cifar10-fast

---

> ### Author Response · Authors · 2022-11-26
> **More experiments, hoping for more discussion**
>
> We finished the Tiny ImageNet experiments that we mentioned in our earlier comment, and they can be found in our paper revision in Appendix B. Like the other new experiments, they show the considerable advantage of using our initialization technique. We've also made other additions, such as a complete implementation, figures showing filters generated using our technique (Appendix A), and experiments demonstrating the impact of our initialization on data efficiency (Appendix B).
>
> You mentioned in your review that you think our method is not really learning free due to the grid search we presented. The Tiny ImageNet results further reinforce the point we made in our earlier comment: the parameters we presented in the paper work for a variety of datasets and architectures, and our initialization can be used "out of the box" without requiring an additional search.
>
> We would appreciate any feedback you have on our new experiments as well as our earlier response. Thanks in advance for your time.

---

> ### Author Response · Authors · 2022-12-05
> **Discussion ends on Dec. 12; we would appreciate hearing back from you**
>
> Hello -- just a reminder that the discussion period will close fairly soon, and we have not yet heard from you regarding our rebuttal and revisions. We have updated the paper and added more experiments, and we think that our results are now even stronger. We have addressed several of the weaknesses that you pointed out:
>
> - Regarding grid search: please see our initial rebuttal for an in-depth explanation of why we think our initialization _really is_ learning-free. But importantly, we added new experiments showing the effectiveness of our initialization on many datasets (CIFAR-100, SVHN, EuroSAT, Tiny ImageNet), where we **used exactly the same parameters as for the CIFAR-10 experiments**. That is, our initialization works on new datasets without requiring a new grid search. To be clear, this is what we expected given the limitations of the grid search (explained in the original rebuttal). But this shows that our technique really **does not require learning.**
>
> - We added several baseline initializations for comparison (orthogonal, delta, zero). To be clear, we didn't include such baselines originally because we had already shown that the technique outperforms the "oracle" initialization of using filters from the pre-trained version of the same model (group D in our graphs on page 5).
>
> - We have added a complete implementation of our initialization.
>
> Thanks in advance for your time and consideration.

---

### Official Review · Reviewer_tQX5 · 2022-11-01

**Confidence:** 4
**Clarity, Quality, Novelty And Reproducibility:** The paper is clear and the ideas and …
**Correctness:** 4
**Technical Novelty And Significance:** 3
**Empirical Novelty And Significance:** 2
**Recommendation:** 8

**Strength And Weaknesses:**

The paper is clear and the motivation / intuition for studying the structure of pre-trained filters is presented well.

Empirical observations are somewhat surprising, showing that transferring from a smaller to a larger network provides more benefits than using covariances computed from the original model.

The main visualizations and comparative results are, unfortunately, based on CIFAR-10 and not on ImageNet. Although some ImageNet results are shown, it would be better if Fig 1, 2 and 3 were based on ImageNet. Some of the main claims from Fig 2, e.g. that transferring from smaller to larger models provides benefits, is also based on what seem to be small improvements (comparing groups C and B).

The proposed initialization seems to be very strongly based on the visualizations of CIFAR-10 trained networks. Although it shows benefits, some comparison against other initialization schemes would be valuable (dirac, orthogonal, zero-init for convs with skip connections, etc).

**Summary Of The Paper:**

The paper studies the structure of depthwise convolutional filters in pre-trained models. Empirically, it shows that computing the covariance of these filters and using it for initialization can bring benefits in terms of trainability and final performance, and such statistics can be transferred from a smaller to a larger model. A new initialization is proposed which further proposes benefits and doesn't require pre-training a network.

**Summary Of The Review:**

The idea is interesting and the visualizations help motivate the approach. The proposed initialization seems a bit too ad hoc and based on heuristics, and, while it yields faster training, the final performance under standard training budgets is similar to the commonly-adopted uniform initialization (bottom rows of Table 1).

---

> ### Author Response · Authors · 2022-11-14
> **New experiments**
>
> Thank you for your review! We have taken your suggestion and added experiments comparing our initialization with dirac, orthogonal, and zero init (as our depthwise convolutions do indeed have skip connections) on CIFAR-10. We also added experiments comparing these initialization techniques to our own on the CIFAR-100, SVHN, and EuroSAT datasets. (And we hope to add more results before the end of the rebuttal period, but wanted to go ahead and start a discussion.) In summary, our technique matches or outperforms all of these methods across the datasets considered. In the "match" cases (SVHN, EuroSAT), we think this is because the task is quite easy and training saturates in relatively few epochs (our method is still better for fewer epochs). We see quite _strong performance gains_ from our technique on CIFAR-100, and it is better on all data sets for short-duration training. Please see Appendix B of our revision for more details.
>
> Our observations of the covariance matrices are indeed mainly based on CIFAR-10-trained models; but the fact of the matter is that our proposed initialization is based only loosely on these covariance matrices anyways, and we show empirically that it often performs very well on ImageNet (and now also CIFAR-100 and more). Our experiments indicated that there may be a way to specify "by hand" the structure of convolutional covariance matrices in a model- and dataset-independent fashion, and we chose the simplest plausible structure we could think of.
>
> We agree it would be great if we could run a similar panel of experiments for ImageNet, but we simply do not have the resources to run hundreds or thousands of such experiments.
>
> While the effect sizes in the graphs, e.g., between groups B and C as you mentioned, may appear very small, looks may be somewhat deceiving (we provided corresponding tables in the appendix to make this clearer). In the 20-epoch setting, the difference between these groups is often around 0.6-1%, and typically 0.2-0.4% for the 200-epoch setting. And note that we ran experiments with three different seeds, so we have a reasonable degree of confidence that these differences are significant. So we think it is surprising and unintuitive that using a smaller model has a significant positive effect on accuracy, even if this effect is "only" around a third of a percent on average.
>
> And in other places (i.e., our final proposed initialization technique), our effect sizes are quite surprisingly large! For example: on ConvMixer-256/24 on CIFAR-10, our initialization does a full 1-2% better than uniform initialization -- and that's when ours is _frozen after initialization_, and the uniform one is actually trained as usual! And **on ImageNet, we also see gains of up to 2% for using our initialization** (thawed setting) -- it would generally take dozens more epochs to reach this performance, which is vastly more expensive than our technique.
>
> You said "the final performance under standard training budgets is similar to the commonly-adopted uniform initialization". While our ~0.1-0.2% gains on full-scale ImageNet training are admittedly rather marginal, please note that our initialization is essentially _completely free_. We would also expect that reasonable initializations eventually converge to the same solution with enough training time (e.g., 150 epochs). And in the CIFAR-10 case, we think our effect sizes are actually quite large for an initialization technique. Please also see our CIFAR-100 results in Appendix B, and the additional convergence plots comparing to other initialization techniques.
>
> Please let us know what you think of the new results; we would be glad to discuss more and to run more suggested experiments.

---

> ### Author Response · Authors · 2022-11-26
> **We ran experiments that you suggested**
>
> We demonstrated that our initialization strongly outperforms **dirac, orthogonal, and zero initializations** as you suggested -- this is in Appendix B of our revised paper. Moreover, we showed that this is the case on a variety of additional datasets (CIFAR-10, CIFAR-100, SVHN, EuroSAT, and Tiny ImageNet).
>
> We hope to hear your opinion on our new results, and also any feedback that you may have about our earlier rebuttal response. Thanks in advance for your time.

---

> ### Author Response · Authors · 2022-12-05
> **Discussion ends on Dec. 12; we would appreciate hearing back from you**
>
> Hello -- just a reminder that the discussion period will close fairly soon, and we haven't heard back from you regarding the updates to our paper or our rebuttal. We have made updates and added new experiments, and believe the paper is now stronger.
>
> In particular, we ran new experiments on new datasets with the **baseline initialization schemes you proposed** (Appendix B).
>
> Further, you commented that our study is too strongly based on CIFAR-10. While we simply don't have the resources to run the same study entirely on ImageNet, our new results show that our initialization is helpful on a variety of different datasets, including "harder" ones than CIFAR-10 (Tiny ImageNet and CIFAR-100). So while many of our observations may have been from CIFAR-10 (and ImageNet, to a lesser extent), the technique we proposed does in fact generalize.
>
> Thanks in advance for your consideration.

---

> > ### Comment · Reviewer_tQX5 · 2022-12-12
> > **Response**
> >
> > Thanks for the response.
> >
> > The additional experiments are very appreciated and help better assess proposed initialization. I have raised my score to reflect this.

---

### Official Review · Reviewer_JJAa · 2022-11-04

**Confidence:** 4
**Correctness:** 3
**Technical Novelty And Significance:** 3
**Empirical Novelty And Significance:** 3
**Recommendation:** 8

**Clarity, Quality, Novelty And Reproducibility:**

- The paper is well written and easy to read.
- I think the main contribution of the paper is the initialization method in section 3. However, I think this contribution is quite modest, given the fact that other techniques exist and the authors don’t compare the different methods.
- Code is not submitted with the paper.

**Strength And Weaknesses:**

***Strengths***:

The initialization method proposed in the paper is interesting and the empirical results are encouraging. The method is very simple to implement.


***Weaknesses***:


**Limited scope:** The experiments in the paper are limited to mainly one architecture called ConvMixer (and also the ConvNeXt architecture). Also, only CIFAR-10 and ImageNet datasets are used. I think it will be interesting to see if the proposed initialization method can work for other architectures and various datasets.

**Comparison to other methods:** The authors mainly compare to the standard uniform initialization (default in PyTorch). I think it will be interesting to compare to other methods (discussed in the related work section). Another relevant work:

Li et. al. FILTER SHAPING FOR CONVOLUTIONAL NEURAL NETWORKS, ICLR 2017

Although other methods are probably more complex, it will be interesting to see the test accuracy that they can achieve, and what is the time complexity of different methods.

**Visual inspection:** I think the authors should better explain what is the structure that we see in Figures 1 and 21. The filters in these figures look rather different, especially in the last layer, although the authors claim that “Covariance matrices from a ConvMixer trained on ImageNet exhibit similar structure to those of ConvMixers trained on CIFAR-10”. For example, the checkerboard structure in figure 21 doesn’t appear in figure 1.
I think the authors should better explain the intuition behind the proposed covariance $\hat{\Sigma}=M \odot (C-0.5S)$. How is the checkerboard structure captured here ?
The authors say that “the sub-blocks of the covariances often have a static negative component in the center, with a dynamic positive component whose position mirrors that of the block itself.” Is it possible that the “dynamic positive component” is the variance, i.e. $[\Sigma_{i,j}]_{\ell,m}$ for $i=\ell,j=m$, and thus it is large ?

**Sensitivity to hyperparameters of the method:** Although the method is quite robust to the hyperparameters of the method (eq. (11)) for CIFAR-10, it is rather sensitive for ImageNet, as we can see in figures 18,19 (and tables 5,6).

***More comments and questions:***

- Can you try to explain or give any intuition why for narrower models 32 filters work better than 256 ? or for shallower models 8 layers better than 32 ?
- The authors don’t tune the hyperparameters of the method for each model and claim that “hyperparameter tuning is optional”. I think it will be interesting to see what can be squeezed from this method by optimal tuning for each model.
- What do you mean by “each data point runs through a full cycle of the LR schedule” in figures 8,11 ?
- In eq. (9): 1/2 is missing
- Will be good to explain how exactly linear interpolation is performed for the shallower models.
- For the shallower models “2- and 4-deep models are also quite effective” – but no results for 2 layers in figure 4.
- In figure 4, for narrower models, there is no comparison to uniform, and no 200 epochs.
- Will be good to explain the “triangular learning rate schedule”.
- I think it will be good to give a quick description of ConvMixer and ConvNeXt architectures.


**Summary Of The Paper:**

In this paper the authors aim to understand the structure of convolution filters, in order to initialize them better. First, the authors estimate the covariance matrix of the filters based on pre-trained models, and sample new filters from a Gaussian distribution with this covariance. Then, based on a visual inspection of the filters in trained networks, the authors propose a simple initialization method. The method seems to work rather well in practice, even when the filters are not trained.

**Summary Of The Review:**

Overall, the paper proposes an interesting method with nice experiments, but I think more work should be done to provide clearer explanations and intuition, with more architectures and comparisons to prior work, before it can be published.

-- POST AUTHOR FEEDBACK:

Thank you for the detailed response and the new results. I am happy to increase my score.

---

> ### Author Response · Authors · 2022-11-14
> **New experiments, clarifications (1/3)**
>
> Thank you for your review. We are still working on our revision to address some of your points and proposed changes to the paper, but we wanted to go ahead and try to start a discussion to clear up any misunderstandings and to present new results.
>
> We have some new experiments, and some more in progress: we compare to suggested **different initialization techniques** (Dirac, zeros, orthogonal), and test our method on **new datasets** (CIFAR-100, SVHN, EuroSAT). **The results uniformly show the strength of our initialization** (please see Appendix B in the updated paper). Our initialization leads to substantial gains on CIFAR-100 compared to baseline initialization techniques, and also smaller improvements on SVHN and EuroSAT. We are working on additional experiments on Tiny ImageNet, and would be glad for suggestions for other datasets/experiments.
>
> We strongly disagree that our contribution is "modest", "given the fact that other techniques exist". We are not aware of any other initialization technique that is _so effective that you can sometimes freeze the weights_, particularly not ones that are _just sampling from a random distribution_. We think it's _quite surprising_ that you can just pull "good enough" convolutional filters from a Gaussian distribution that was built in some sense "by hand" -- and moreover, that this distribution works (without changing the hyperparameters) on a variety of datasets and architecture configurations.
>
> Also, our initialization often leads to gains of an _entire percent_ or more, despite the fact that it is, again, is _just blindly sampling from a Gaussian_. On a deep model (ConvMixer-256/24), our filter initialization is so effective that you can gain _1-2% accuracy_ relative to uniform even if you _don't train the filters_. Our initialization helps achieve more accuracy in less time on a variety of datasets (which is bolstered by our new results), and it _costs nothing_.
>
> Further, deep models using our initialization can gain multiple percents of accuracy on ImageNet _for free_; an additional 2% of accuracy with ConvMixer-1024/32 could take dozens more epochs, which is vastly more expensive than using our technique. While the difference fades as the number of epochs increases, we think this is what anyone would expect of a reasonable initialization technique.
>
> While our paper is limited in scope to new CNNs that use large-filter depthwise convolutions, the two architectures we studied are quite widely-used.
>
>
> ### Related work
>
> First, we want to point out that we are not aware of any strongly-related work that we have not already covered in the paper. The work you linked, "FILTER SHAPING FOR CONVOLUTIONAL NEURAL NETWORKS", contains similar _keywords_ to our paper, but we don't think it is particularly salient. Their method shapes filters by _analyzing the dataset_. They discuss the covariance _of the dataset_ and how it can influence the choice of filter shape, while we are concerned with the covariance _of the filters themselves_. Our work deals with performance on natural images, which differs considerably from the datasets used in the paper; they explicitly mention in the paper that square convolutions (like ours) are appropriate for natural images and/or domains where we have an abundance of training examples. Their method adds an actual _constraint_ on the shape of the filters; we present an _initialization_ that happens to put more _probability density_ in the center of the filter. These are very different things -- ours is, after all, just a random initialization technique. In any case, we could not find resources to easily replicate the work for a comparison -- though it's unclear what the comparison would be, as the filter shape they propose for natural images is square, the same as ours.
>
> And to be clear, we think that our latest results make it (even more) clear that our technique is _not dependent on the dataset_.
>
> You asked "what is the time complexity of different methods", but this conveys a misunderstanding of our work. Our method simply involves sampling from a (relatively low-dimensional) multivariate Gaussian distribution; the compute requirements are not a concern at all, similarly to uniform initialization. While we are aware of initialization techniques that involve non-trivial pre-computation, we purposefully compare to none of these, because we consider them to be fundamentally different from our direction.
>
> We didn't present comparisons to other common initializations (orthogonal, dirac/identity, etc.) in the paper because we already compared to an "oracle" initialization: we trained the model first for 200 epochs, and then used those filters _directly_ from the trained model to initialize a new one (group D in our Figure 3, 4 graphs). Even in the frozen setting, our initialization outperforms this strategy, which we find quite interesting.

---

> > ### Author Response · Authors · 2022-11-14
> > **New experiments, clarifications (2/3)**
> >
> > ### Visual inspection
> >
> > We acknowledge that the structures seen for CIFAR-10 and ImageNet are quite different. We didn't try to capture all aspects of this structure, but rather the ones we considered most significant -- that is, we built a relatively simple class of covariance matrices loosely inspired by these visual observations. Despite the fact that our initialization only captures some of the structure we observed, our experiments indicate that it is very effective (which is what we consider to be important).
> >
> > We also think we adequately explained the justification for $\hat{\Sigma} = M \odot (C - \tfrac{1}{2} S)$, in that we broke down the structure into three components we observed (static, dynamic, mask) and explained how these components map to our model (S, C, M respectively). We didn't attempt to capture the checkerboard structure (nor did we claim to). Please let us know if something in particular was unclear about this, but we are not sure how to improve at the moment.
> >
> > Yes, this is the variance of pixel i, j in the filter.
> >
> > ### Sensitivity
> >
> > Our graphs here are not as clear as they could be: The color scale exaggerates what are actually fairly small differences (e.g., 55.3% and 55.4% look quite dramatically different in the middle graph for thawed filters). Also, note that the CIFAR-10 grids were averaged over 3 trials each, while we could only afford 1 trial for ImageNet. We also only ran for 10 epochs, compared to 20 epochs for CIFAR-10 (a _much_ smaller dataset), so it's not surprising that there is more variance in the results. While this is not ideal, it's subject to our compute constraints (and we still believe the graphs don't show dramatic variance w.r.t. hyperparameters).
> >
> > Furthermore, we reported ImageNet results on two very different configurations (best for frozen vs. best for thawed, "acceleration" 0.25 vs. 1.0), and we generally did not see very large accuracy differences between the two for 50-epoch training.

---

> > > ### Author Response · Authors · 2022-11-14
> > > **New experiments, clarifications (3/3)**
> > >
> > >
> > > ### More comments and questions
> > >
> > > * We're not really sure why smaller models tend to work better here. One observation is that the covariance matrices from smaller models are noisier in some sense, despite exhibiting the same structure. So, we thought that more noise in the covariance matrix might be beneficial for the initialization, and we have tried making the covariance from same-sized models (as well as our Kronecker-factor modeling experiment in the appendix) noisier to attempt to recreate the effect. We tried both adding random Gaussian noise to the covariance matrix (the whole matrix or just the diagonal) and also resampling it, but there was no clear benefit to either. It may be that it's particularly hard to learn filters for deeper models/they take longer to train, so that shallower models have learned "better" filters that we can simply interpolate. But we don't have good intuition for the narrower case.
> > >
> > > * This would certainly be interesting -- we would like to see if the optimal hyperparameters of our initialization correlate with certain architecture decisions, e.g., patch size. However, we simply do not have the compute to tune our method for every model (and it's unclear how many epochs we would actually have to use for each trial to make this experiment meaningful).
> > >
> > > * The learning rate schedule is triangular, i.e., it increases, peaks mid-training, and then decreases -- and this is compressed or expanded to fit the total number of iterations. We are trying to say that each point in these graphs is an entire training run with $x$ epochs -- the graphs _are not_ individual training runs (i.e., they are not the loss curves more often seen). Instead, we're asking the question, "we want to get maximal performance out of $x$ epochs, how well can we do with each initialization?". Generally, having an LR schedule tailored to the number of epochs like this is quite a bit more effective than, say, stopping a 100-epoch training schedule prematurely at epoch $x$.
> > >
> > > * Ah, thanks, we will fix this.
> > >
> > > * We'll add this to the appendix. Basically, as with our initialization, we map layers to their "percent depth", which is between 0 and 1. Then any covariance in the target model is simply the weighted sum of the two covariances it is "between" in the base model. To be completely clear, we have added the Python function we implemented for this to the appendix.
> > >
> > > * Oops, we'll change this to just 4-deep models.
> > >
> > > * In this graph, we more so wanted to show the trend (which increases and then decreases), rather than make any statement about being better/worse than another initialization technique. We thought that the 20- and 50- epoch results were sufficient to see this trend (and generally, all of our initializations have more significant effects for short training). Currently, we are trying to run more experiments on different initialization techniques and datasets, so fixing this seems fairly low priority (we have a limited compute budget, so we cannot do both at the same time). But we will try to get around to this.
> > >
> > > * Sure, we added details on the LR schedule in the appendix (A.1), and will add details on the architectures as well. We will also make the general structure of these architectures more clear in the main text, e.g., in "Scope": they both essentially involve patch embeddings followed by a stack of depthwise + pointwise convolutions.
> > >
> > > Thank you again for your review, and we look forward to hopefully discussing more.

---

> > > > ### Comment · Reviewer_JJAa · 2022-11-19
> > > > **Response**
> > > >
> > > > Thank you for the detailed response.
> > > >
> > > > **Experiments:**
> > > >
> > > > The new experimental results with more datasets look encouraging and strengthen the paper. Will be good to try also more architectures with depthwise convolutions.
> > > >
> > > > **Time complexity:**
> > > >
> > > > I didn’t try to claim that time complexity is a disadvantage of your method. I was hoping that by comparing time complexity vs test accuracy of different initialization methods you can show how your method compares to other methods.
> > > >
> > > > **Visual inspection:**
> > > >
> > > > I am rather confused by the authors saying in the paper that “Filter covariance matrices in pre-trained ConvMixers and ConvNeXts have a great deal of structure, which we observe across models with different patch sizes, architectures, and data sets”…”we noticed clear repetitive structure”. However, the authors say here that “we built a relatively simple class of covariance matrices loosely inspired by these visual observations”. So, if you are “loosely inspired by these visual observations”, is there a clear structure ? The title of the paper is “UNDERSTANDING THE COVARIANCE STRUCTURE..”, but even for 2 datasets (CIFAR-10 and ImageNet) we see a different structure (Figure 1 vs 30), so what do we “understand” ?
> > > >
> > > > In your method you try to capture the static part and the dynamic part. The dynamic part is trivial – it is just the variance, which moves according to pixel index $(i,j)$. I think the authors should clearly state it in the paper, otherwise, the reader can think that this part is surprising. Can you think of a simple explanation for the static part ? Is it because many of the filter entries are just far from the center of the filter, and thus the correlation is small (or negative) ?
> > > >
> > > > I think it will be good to show a few examples of how the filters (sampled from Gaussian distribution with your covariance) look like. Are they similar to Gaussian filters ? Anyway, I think the authors should compare their method against this simple baseline: just initialize all filters with a Gaussian filter (with a fixed variance, or increase the variance like in Eq. (11)). In this way we can see if the added complexity of building the covariance and sampling is really needed.

---

> > > > > ### Author Response · Authors · 2022-11-21
> > > > > **We ran your proposed baseline; more discussion (1/2)**
> > > > >
> > > > >
> > > > > ## Experiments
> > > > >
> > > > > We have added some new experiments on Tiny ImageNet (our initialization also does better than the baselines), as well as some showing the effect of our init on data efficiency on CIFAR-10 (our init is particularly advantageous in the low-data setting).
> > > > >
> > > > > Which architectures with depthwise convolutions do you have in mind? Our method is meant for large-filter depthwise convolutions, which have only become popular recently -- this was popularized by ConvMixer and ConvNeXt, the two architectures we use in the paper.
> > > > >
> > > > > ## Time complexity
> > > > >
> > > > > Do you have another initialization method in mind? We think it’s really only fair to compare our technique to initializations that are also "compute-free". There would be no reason to compare time complexities with the baselines we just added to the appendix (Dirac, zeros, orthogonal, uniform) -- generating the weights with these techniques takes practically no compute, just like our technique.
> > > > >
> > > > >
> > > > > ## Visual inspection
> > > > >
> > > > > ### Example filters from our technique
> > > > > As you suggested, we added some plots showing more covariance matrices, as well as filters generated from those covariance matrices and the corresponding filters generated from our own technique. This is **Figure 13 in Appendix A**. To answer your question, the filters generated from our technique are not really similar to Gaussian filters.
> > > > >
> > > > > ### Gaussian filter baseline
> > > > >
> > > > > We also ran the simple baseline you suggested -- we initialized all filters with a Gaussian filter, increasing the variance as in Eq. 11. We used the same hyperparameters as for all our CIFAR/SVHN/EuroSat/Tiny ImageNet experiments. We report the results below since we can no longer update the paper.
> > > > >
> > > > > Results for _your_ proposed technique, _our_ technique, and uniform initialization on a ConvMixer-256/8 with patch size 2:
> > > > >
> > > > > | Ks | Epochs | | | Thawed | | | | Frozen | |
> > > > > |-|-|-|-|-|-|-|-|-|-|
> > > > > | | |  | **Uniform** | **Ours** | **Yours** |  | **Uniform** | **Ours** | **Yours** |
> > > > > | 7 | 20 |  | 89.66 $\pm$ 0.21 | 90.79 $\pm$ 0.24 | 89.57 $\pm$ 0.11 |  | 86.73 $\pm$ 0.27 | 90.22 $\pm$ 0.03 | 72.81 $\pm$ 0.17 |
> > > > > | 7 | 50 |  | 91.81 $\pm$ 0.15 | 92.48 $\pm$ 0.08 | 91.61 $\pm$ 0.16 |  | 89.44 $\pm$ 0.23 | 92.02 $\pm$ 0.23 | 78.58 $\pm$ 0.18 |
> > > > > | 7 | 200 |  | 92.86 $\pm$ 0.15 | 93.40 $\pm$ 0.20 | 92.76  $\pm$ 0.32 |  | 90.70 $\pm$ 0.12 | 92.83 $\pm$ 0.24 | 81.25 $\pm$ 0.28 |
> > > > > |  |  |  |  |  |  |  |  |  |  |
> > > > > | 9 | 20 |  | 89.26 $\pm$ 0.35 | 90.85 $\pm$ 0.14 | 89.26 $\pm$ 0.25 |  | 84.97 $\pm$ 0.03 | 90.56 $\pm$ 0.09 | 73.22 $\pm$ 0.53 |
> > > > > | 9 | 50 |  | 91.74 $\pm$ 0.12 | 92.54 $\pm$ 0.16 | 91.59 $\pm$ 0.26 |  | 88.25 $\pm$ 0.39 | 91.96 $\pm$ 0.12 | 78.38 $\pm$ 0.16 |
> > > > > | 9 | 200 |  | 92.65 $\pm$ 0.16 | 93.21 $\pm$ 0.09 | 92.66 $\pm$ 0.14 |  | 90.04 $\pm$ 0.33 | 93.00 $\pm$ 0.05 | 81.15 $\pm$ 0.10 |
> > > > >
> > > > > As you can see, your technique typically does about the same or worse than uniform initialization (with thawed filters). With frozen filters, yours does substantially worse than both techniques. In all cases, our technique does substantially better than yours and uniform. So we really do need to build the covariance and sample from it.
> > > > >
> > > > > Have you considered our results when filters are frozen after initialization? This showcases the unusual effectiveness of our initialization.

---

> > > > > > ### Author Response · Authors · 2022-11-21
> > > > > > **We ran your proposed baseline; more discussion (2/2)**
> > > > > >
> > > > > > ### Dynamic part is "trivial"?
> > > > > >
> > > > > > The dynamic part is not trivial: it is _not just_ the variance. While location i, j within block i, j represents the variance of pixel i, j, that’s not the entirety of the "dynamic part". For one, the variance is dependent on the location of the pixel, with pixels further from the center having less variance (already a large difference from the univariate distributions typically used for initialization) -- thus the mask that interacts with the dynamic part. For two, the dynamic part _isn’t a single pixel_ -- it includes the neighbors of the pixel -- in our model, it’s a Gaussian centered at i, j. So filter pixels are correlated with their neighbors depending on proximity, and this correlation also decays as you move away from the center of the filter.
> > > > > >
> > > > > > Is this _surprising_? It’s up to the reader. This seems like quite a _reasonable_ construction of the covariance of convolutional filters (that is, after all, what we wanted!). But is this _trivial_? I seriously do not think this is trivial.
> > > > > >
> > > > > > We think the static component may make the filters generated from the distribution more "edge-detector-like", i.e., ensuring that they have juxtaposed positive and negative entries, or a well-defined "high" and "low" component. We think this is the simplest way to produce such an effect. Through  _lots_ of experiments, we have shown that this construction is very effective. And we have seen in empirical covariance matrices that this static component is typically present.
> > > > > >
> > > > > > ### What did we really "understand"?
> > > > > >
> > > > > > When we said “clear structure”, we didn’t mean, “we can clearly write down an equation that perfectly describes that exact structure”. We meant that the covariances _clearly_ have _some sort of_ structure: they aren’t just random noise, they aren’t just identity matrices -- do you disagree that the covariance matrices seem to have structure? As opposed to, say, random noise? We do see repetitive structure, quite literally: look at Layer #1 of 8 in Figure 13, and look at Layer #7 of 12 in Figure 31, and look at Layer #1 of 8 in Figure 1. Do they not look similar to you? It’s perfectly possible to acknowledge that a pattern or structure _exists_, without being able to perfectly describe it mathematically.
> > > > > >
> > > > > > **And we have understood lots about the covariance structure of convolutional filters.**
> > > > > >
> > > > > > 1. The covariance from learned models (not the weights themselves!) can effectively initialize new models -- it was not a priori obvious that the structure would have such stability across different models / training runs
> > > > > > 2. The covariance transfers across different architecture configurations such as width, depth, patch size, and kernel size -- this suggests that the structure  may be "parameterized", insofar as we can smoothly interpolate it across some of these variables
> > > > > > 3. The covariance of learned filters has, to some degree, Kronecker product structure (Appendix A), which is an additional datapoint that it is compressible / amenable to modeling
> > > > > > 4. We propose a simple model of the covariance structure in learned filters, and demonstrate its effectiveness relative to the "true" structure (it's typically even better) -- which suggests we have captured an important part of the structure (even if not exactly _all_ of it).
> > > > > >
> > > > > > Our goal in understanding the structure was to propose an initialization technique -- and we have shown through _very many_ experiments that our initialization technique is _very effective_, which makes us believe that we have captured (and thus understood) some of the important structure of the covariance matrices of convolutional filters.
> > > > > >
> > > > > > Again -- filters sampled from the Gaussian defined by our covariance construction are so effective, that in many cases they don’t really have to be trained. Sometimes, they’re even better than models using filters that were trained end-to-end! This applies to CIFAR-10, ImageNet, Tiny ImageNet, and EuroSAT, and to a lesser extent CIFAR-100 and SVHN. So while CIFAR-10 was a useful case study to propose our initialization, our technique generalizes beyond it (more evidence we have understood the structure). The covariance structure we designed _works really well_ -- and it’s entirely closed-form, and it can be interpreted in intuitive terms -- how does that not constitute understanding the covariance structure of convolutional filters? The fact that we can often freeze filters after initialization suggests that we have a closed-form description of a large chunk of the weights of practical neural networks (and thus some understanding); yes, this doesn't work as well as training in _all_ cases, but we think this is quite surprising.

---

> > > > > > > ### Comment · Reviewer_JJAa · 2022-11-21
> > > > > > > **Response**
> > > > > > >
> > > > > > > Thank you again for the detailed feedback.
> > > > > > >
> > > > > > > I think the paper can benefit from a more accurate description of what is the dynamic part, following the discussion here.
> > > > > > >
> > > > > > > In addition, I would like to propose performing some ablation study to understand the contribution of each part. For example, if you don't model the static part, which filters are missing (edge-detector?), etc..
> > > > > > > This ablation study can give a better understanding of why/if all parts of your model are important.
> > > > > > >
> > > > > > > I am also happy to raise my score to 6.

---

> > > > > > > > ### Author Response · Authors · 2022-12-06
> > > > > > > > **Ablation study (1/2)**
> > > > > > > >
> > > > > > > > We were able to get some results for the ablation study you proposed. The difficulty with this is that you can't really ablate _just_ the static part, because it will re-appear in the symmetric projection since $S^B = M$, as mentioned below Eq. 7 on page 6. And we also can't avoid the symmetric projection: covariance matrices _have_ to be symmetric. If we left out this step, it would just be implemented implicitly by, e.g., `np.random.multivariate_normal`, which projects the covariance matrix argument to the closest symmetric PSD matrix. Nonetheless, we have tried a few different combinations, which we'll explain below... in order to make the table compact, we've labeled each option as A, B, C, D, or E.
> > > > > > > >
> > > > > > > > ### A (The original model)
> > > > > > > >
> > > > > > > > This is just the original model, i.e., $M \odot (C - \tfrac{1}{2} S)$. As shown in Eq. 10, the symmetric projection of this is $\tfrac{1}{2} ( M \odot (C - S) + S \odot C)$.
> > > > > > > >
> > > > > > > > ### B (Attempting to ablate S)
> > > > > > > >
> > > > > > > > Here we remove the static component $S$ from the model, getting $M \odot C$. But the static component reoccurs after the symmetric projection: $\tfrac{1}{2} ( M \odot C + (M \odot C)^B ) = \tfrac{1}{2} ( M \odot C + S \odot C) = \tfrac{1}{2} C \odot  (M + S)$.
> > > > > > > >
> > > > > > > > ### C (Attempting to ablate C)
> > > > > > > >
> > > > > > > > If we remove the dynamic component $C$ from the model, we have just $\tfrac{1}{2} M \odot S$, and it's easy to see the symmetric projection is the same.
> > > > > > > >
> > > > > > > > ### D (Attempting to ablate M)
> > > > > > > >
> > > > > > > > After removing M, we get $C - \tfrac{1}{2}S$, and after the symmetric projection this is $\tfrac{1}{2} ( C - \tfrac{1}{2}S + (C - \tfrac{1}{2}S)^B) = \tfrac{1}{2}(2C - \tfrac{1}{2}S - \tfrac{1}{2}M) = C - \tfrac{1}{4} S - \tfrac{1}{4}M$.
> > > > > > > >
> > > > > > > > ### E (Ablating both M and S)
> > > > > > > >
> > > > > > > > The only way to really get rid of the static component is to get rid of both $M$ and $S$, which gives us just $C$. Since $C$ is symmetric, the final model is just $C$ itself.
> > > > > > > >
> > > > > > > > --------------
> > > > > > > >
> > > > > > > > First, we visually inspected filters drawn from these models.
> > > > > > > >
> > > > > > > > - **B** is pretty similar to **A**. The difference between "high" and "low" is a bit less dramatic, but they look quite similar.
> > > > > > > > - **C** looks like Gaussian filters with random variance and scale, i.e., there are no edge-detector-like features without the inclusion of the mask or static component. These look very different from our model, and more like your earlier Gaussian-filter baseline.
> > > > > > > > - **D** and **E** essentially look like noise for low settings of the variance, but they acquire some structure as this parameter increases. These also look very different.
> > > > > > > >
> > > > > > > >
> > > > > > > > We tried out these different configurations with a ConvMixer-256/8 with 2x2 patches on CIFAR-10.
> > > > > > > >
> > > > > > > > | |  |  |  | Thawed |  |  |  |  | Frozen |  | |
> > > > > > > > |---|---|---|---|---|---|---|---|---|---|---|---|
> > > > > > > > |Ks | Epochs | **A** | **B** | **C** | **D** | **E** | **A** | **B** | **C** | **D** | **E**|
> > > > > > > > |7|20| **90.86 $\pm$ .13** |90.78 $\pm$ .14|89.80 $\pm$ .18|88.05 $\pm$ .24|87.89 $\pm$ .26|**90.28 $\pm$ .11**|90.18 $\pm$ .16|77.95 $\pm$ .28|86.28 $\pm$ .20|86.17 $\pm$ .17|
> > > > > > > > |7|50| **92.52 $\pm$ .18**|92.46 $\pm$ .12|91.65 $\pm$ .21|90.77 $\pm$ .13|90.71 $\pm$ .23|91.87 $\pm$ .22|**91.87 $\pm$ .14**|82.92 $\pm$ .23|89.20 $\pm$ .20|88.95 $\pm$ .22|
> > > > > > > > |7|200|**93.29 $\pm$ .15**|93.17 $\pm$ .18|92.47 $\pm$ .20|92.31 $\pm$ .11|92.26 $\pm$ .12|**92.80 $\pm$ .07**|92.62 $\pm$ .08|85.41 $\pm$ .31|90.90 $\pm$ .09|90.66 $\pm$ .24|
> > > > > > > > |9|20|90.73 $\pm$ .25|**90.88 $\pm$ .26**|89.86 $\pm$ .25|86.64 $\pm$ .36|86.17 $\pm$ .06|**90.49 $\pm$ .05**|90.13 $\pm$ .14|78.36 $\pm$ .36|84.80 $\pm$ .26|84.39 $\pm$ .30|
> > > > > > > > |9|50|**92.63 $\pm$ .09**|92.38 $\pm$ .13|91.48 $\pm$ .07|90.01 $\pm$ .40|89.83 $\pm$ .08|91.97 $\pm$ .07|**91.98 $\pm$ .22**|82.79 $\pm$ .10|87.69 $\pm$ .23|87.54 $\pm$ .07|
> > > > > > > > |9|200|**93.27 $\pm$ .12**|93.15 $\pm$ .29|92.56 $\pm$ .13|92.05 $\pm$ .27|92.02 $\pm$ .15|**92.88 $\pm$ .21**|92.76 $\pm$ .06|85.15 $\pm$ .12|89.90 $\pm$ .46|89.74 $\pm$ .08|
> > > > > > > >
> > > > > > > > Our results are in line with our visual observations. **C**, **D**, and **E** simply do not work very well, which indicates that we really need the combination of all three components for the initialization to work well. However, **B** does almost as well as **A**, and we can't say with _much_ confidence that the two are statistically significantly different. **A**, the original model, has a slight advantage in some experiments, and is also more in line with our visual observations and/or more intuitive. **B** also does slightly better in some experiments, though again, error bars are often overlapping. **B** has a shorter description length, but **A** has a visible "negative" static component $-M\odot S$ like we observed empirically.

---

> > > > > > > > > ### Author Response · Authors · 2022-12-06
> > > > > > > > > **Ablation study (2/2)**
> > > > > > > > >
> > > > > > > > > What we _can_ say is that we do, in fact, need all three components -- and, unlike in group **D**, they need to "interact" with each other.
> > > > > > > > >
> > > > > > > > > As a sort of tiebreaker between **A** and **B**, we were able to run some experiments using a model with 1x1 patches:
> > > > > > > > >
> > > > > > > > > | |  |  | Thawed |  | Frozen|
> > > > > > > > > |---|---|---|---|---|---|
> > > > > > > > > |Ks | Epochs | **A** | **B** | **A** | **B**|
> > > > > > > > > |9|20|**92.31 $\pm$ .18**|92.14 $\pm$ .23|**91.02 $\pm$ .21**|90.63 $\pm$ .10|
> > > > > > > > > |9|50|**93.43 $\pm$ .16**|93.26 $\pm$ .13|**92.25 $\pm$ .16**|92.02 $\pm$ .15|
> > > > > > > > > |9|200|**93.87 $\pm$ .15**|93.63 $\pm$ .12|**93.06 $\pm$ .29**|92.94 $\pm$ .17|
> > > > > > > > >
> > > > > > > > > The advantage of **A** over **B** is more well-defined here. It's worth noting that **A**'s accuracy dominates **B**'s over the whole course of training -- and the table is not the best presentation, as **A**'s accuracy was higher than **B**'s in all trials. Ultimately, we think the original model is the right one, and it usually does better than the very-similar **B**-initialization.
> > > > > > > > >
> > > > > > > > > ----------------
> > > > > > > > >
> > > > > > > > > As for making the dynamic part clearer, we propose the following changes:
> > > > > > > > >
> > > > > > > > > - State that C, S, and M are the dynamic, static, and mask components respectively in the caption of Figure 5 -- we think the graphics make the nature of the components very clear, but maybe listing the variable names is not as ideal as calling them the dynamic, static, and mask components.
> > > > > > > > > - We will refer to Fig. 1 in "Visual Observations" in Sec. 3, and then explicitly point out that the components are not comprised of points within blocks, but rather Gaussian-like densities centered at those points. This should also make the transition to talking about modeling the covariances in terms of Gaussian filters a bit smoother.
> > > > > > > > >
> > > > > > > > > ------------
> > > > > > > > >
> > > > > > > > > Thanks again for your time and for engaging with the paper.

---

> > > > > > > > > > ### Comment · Reviewer_JJAa · 2022-12-07
> > > > > > > > > > **Nice**
> > > > > > > > > >
> > > > > > > > > > Thank you for these new results.
> > > > > > > > > >
> > > > > > > > > > I am happy to raise my score to 8.

---

### Decision · Program_Chairs · 2023-01-20

**Decision:**

Accept: poster

**Justification For Why Not Higher Score:**

The suggested method and empirical observations are nice, but do not seem ground-breaking.

**Justification For Why Not Lower Score:**

All reviewers recommended accept [8,8,6,6] and there are no major concerns remaining.

**Metareview: Summary, Strengths And Weaknesses:**

The paper studies the structure of depthwise convolutional filters in pre-trained ConvMixer and ConvNeXt models, which have large kernels. It shows empirically that computing the covariance of these filters and using it for initialization can improve the trainability and final performance, and that such statistics can be transferred from a smaller to a larger model. Based on this, a new initialization is proposed which doesn't require pre-training a network, and improves over several baselines. Interestingly, sometimes, this improvement can be achieved without training the depthwise convolutional filters at all. This empirical observations in this paper seem novel and interesting, and the suggested method seems practically useful.

The reviewers had some concerns, but these were mostly addressed during the discussion period, e.g. with some additional experiments and ablation studies.

**Note From Pc:**

if the above contains the word "oral" or "spotlight" please see: "oral" presentation means -> notable-top-5% and "spotlight" means -> notable-top-25%. As stated in our emails, we are disassociating presentation type from AC recommendations